# POLIPHON conversion factors for retrieving dust-related cloud condensation nuclei and ice-nucleating particle concentration profiles at oceanic sites

Yun He[1,2,3,*], Zhenping Yin[4,*], Albert Ansmann[5], Fuchao Liu[1,2,3], Longlong Wang[4], Dongzhe Jing[1,2,3],
Huijia Shen[1]

[1]School of Electronic Information, Wuhan University, Wuhan, China
[2] Key Laboratory of Geospace Environment and Geodesy, Ministry of Education, Wuhan, China
[3]State Observatory for Atmospheric Remote Sensing, Wuhan, China.
[4]School of Remote Sensing and Information Engineering, Wuhan University, Wuhan, China.
[5]Leibniz Institute for Tropospheric Research, Leipzig, Germany

*Correspondence to*: Yun He (heyun@whu.edu.cn), Zhenping Yin (zp.yin@whu.edu.cn)

**Abstract.** Aerosol-cloud interactions (ACI) are the largest contributor to the uncertainty in the global radiation budget. To improve the current consideration of ACI in global circulation models, it is necessary to characterize the 3-D distribution of
dust-related cloud condensation nuclei concentration (CCNC) and ice nucleating particle concentration (INPC) globally. This can potentially be realized using the POLIPHON (POlarization LIdar PHOtometer Networking) method together with spaceborne lidar observations. However, dust-related conversion factors that convert bulk aerosol optical properties from lidar measurements to aerosol microphysical properties, are still less constrained in many regions, which limits the applications of the POLIPHON method. Here we retrieve the essential dust-related conversion factors at remote
oceanic/coast sites using the historical AERONET (AErosol RObotic NETwork) database. Depolarization-ratio-based dust ratios $R_d$ at 1020 nm are applied to identify the dust-occurring cases, thus enabling us to contain fine-mode dust dominated cases (after the preferential removal of large-size dust particles during transport), study the evolution of dust microphysical properties along the transoceanic pathway, and mitigate occasional interference of large-size marine aerosols. The newly proposed scheme is proven to be valid and feasible by intercomparisons with previous studies at nine sites in/near deserts.
The dust-related conversion factors are calculated at 20 oceanic/coast sites using both PD (pure dust) and PD+DDM (dust-dominated mixture) datasets. At nearly half of the sites, the conversion factors are solely calculated using the PD datasets, while at the remaining sites, the participation of DDM datasets is required to ensure a sufficient amount of data for the calculation. Evident variation trends in conversion factors are found for $c_{v,d}$ (extinction-to-volume, gradually decreasing), $c_{250,d}$ (extinction-to-particle (with radius >250 nm) number concentration, gradually increasing) and $c_{s,d}$ (extinction-to-
surface area concentration, gradually decreasing) along both the transpacific and transatlantic dust transport pathways. The retrieved dust-related conversion factors are anticipated to inverse 3-D dust-related CCNC and INPC distributions globally, thereby improving the understanding of ACI in atmospheric circulation models.

## 1 Introduction

Clouds are widely present in the Earth's atmosphere, as they cover approximately 2/3 of the Earth's surface. They play an essential role in weather, hydrology, climate, air chemistry, and several practical applications (Spänkuch et al., 2022). Clouds modify the radiation budget of the Earth by regulating the incoming solar radiation and outgoing longwave radiation, thus significantly affecting the global climate. The level of induced net radiation is strongly associated with the microphysical characteristics of cloud particles (either cloud drops or ice crystals), i.e., size, concentration, phase, and shape. These characteristics are greatly influenced by aerosol-cloud interactions (ACI), also known as 'aerosol indirect effects', thus resulting in the largest uncertainties in the global effective radiative forcing (IPCC, 2021; Rosenfeld et al., 2014).

For liquid-water clouds, aerosols serve as cloud condensation nuclei (CCN) to alter the drop size and albedo (Twomey, 1974) as well as to postpone the initiation of rainfall and increase the lifetime and coverage of clouds (Albrecht, 1989). For mixed-phase and ice clouds, heterogeneous nucleation is another crucial effect in which proper aerosol particles may act as ice-nucleating particles (INP) to trigger in-cloud ice formation (from either vapor or liquid water) (Ansmann et al., 2019a, 2019b, 2021; Murray et al., 2012; Cziczo et al., 2013; Yin et al., 2021). In the study of ACI, it is of great importance to quantify the concentration of CCN and INP. In recent years, aerosol optical depth (AOD) and aerosol index (AI) have been found to be inaccurate proxies for CCN (Shinozuka et al., 2015; Stier, 2016), thus motivating the scientific community to estimate the CCN concentration (CCNC) (Mamouri and Ansmann, 2016; Georgoulias et al., 2020; Choudhury and Tesche, 2022a, 2022b; Patel et al., 2022; Lenhardt et al., 2022). Phillips et al. (2013) stated that the reliable quantification of the linkage between aerosol conditions and ice crystal numbers should be the first step in quantifying cold-cloud indirect effects. Thus, INP concentrations (INPC) are estimated in many studies to predict the initial in-cloud ice crystal number concentration (ICNC) via primary heterogeneous nucleation (Ansmann et al., 2019a; He et al., 2022a; Kanji et al., 2017). Moreover, discrepancies between INPC and ICNC are found to establish the role of secondary ice nucleation (DeMott et al., 2011). Therefore, the constraint of ambient INPC can lead to an accurate representation of cloud microphysical processes and reduce the uncertainties in estimating the climate feedback associated with ice and mixed-phase clouds in climate models (Li et al., 2022).

To estimate the CCNC and INPC, the POLIPHON (POlarization LIdar PHOtometer Networking) method was developed by Ansmann et al. (2012) and has been improved for years in several field campaigns (Mamouri and Ansmann, 2014, 2015, 2016, 2017). As a remote sensing approach, it has been well verified through comparisons with simultaneous in-situ measurements (Marinou et al., 2019; Wieder et al., 2022); hence, it is applicable for the analysis of long-term observations. The POLIPHON method has been proven to be useful for examining the profiles of CCNC, INPC, and aerosol number concentration retrieved from spaceborne lidar measurements with other algorithms (Choudhury and Tesche, 2022a, 2022b; Choudhury et al., 2022c). In the POLIPHON method, the lidar-derived aerosol extinction coefficient profiles are first divided

into the respective contributions from different aerosol types (Tesche et al., 2009; Mamouri and Ansmann, 2016, 2017) as well as from fine and coarse mode components (Mamouri and Ansmann, 2014). Then, these aerosol-type-dependent extinction coefficient profiles are converted into CCNC and INPC profiles by employing the photometer-data-derived conversion factors and different CCN and INP parameterizations. Conversion factors connect the lidar-retrieved aerosol optical properties and aerosol microphysical properties and thus can be used to estimate CCNC and INPC straightforwardly.

Dust aerosols are the most abundant aerosol type by mass in the atmosphere (Kok et al., 2021a) and are considered both effective CCN and INP (Mamouri and Ansmann, 2016; He et al., 2021a, 2021b, 2022a, 2022b; Murray et al., 2012; Kanji et al., 2017). Therefore, dust-related CCNC and INPC estimations should be receive special attention. Numerous field campaigns have been conducted to study regional CCNC and INPC profiles (Haarig et al., 2019; Hofer et al., 2020; Engelmann et al., 2021; He et al., 2021b). Using spaceborne lidar, e.g., CALIOP (Cloud-Aerosol Lidar with Orthogonal

Polarization) launched in April 2006 (Winker et al., 2007) and the currently ongoing EarthCARE mission (Illingworth et al., 2015), we can characterize the 3-D distribution of dust-related CCNC and INPC at a global scale. Therefore, the number concentration of cloud drops and ice crystals can be well quantified (Ramanathan et al., 2001), thus improving the current consideration of ACIs in global circulation models (Mamouri and Ansmann, 2015; Froyd et al., 2022).

To do this, the first challenge is to retrieve the global dust-related conversion factors. However, they are generally

regionally variable and dependent on the microphysical properties of dust particles. Mamouri and Ansmann (2015) first calculated the dust-related conversion factors based on sun/sky photometer data during several field campaigns, including SAMUM-1 (Saharan Mineral Dust Experiment), SAMUM-2, SALTRACE (Saharan Aerosol Long-range Transport and Aerosol–Cloud Interaction Experiment), and long-term observations in Limassol. Subsequently, Ansmann et al. (2019b) obtained dust-related conversion factors using AERONET (AErosol Robotic NETwork) data at typical sites near the main

deserts. They assumed the predominant contribution of dust in the atmospheric column and systematically applied the column-integrated Ångström exponent (AE for 440–870 nm) < 0.3 and AOD at 532 nm > 0.1 as the criteria in selecting dust presence cases. Dust particles are frequently elevated from the surface of desert regions by wind or thermal convection and can sometimes undergo advective transport over a long range. The dust-related conversion factors can be very different for the downstream areas far from the dust sources due to the possible aging and mixing of dust with other aerosol types during

long-range transport (Kim and Park, 2012; Goel et al., 2020). However, for these downstream areas, dust-related conversion factors are still lacking owing to insufficient data points fulfilling the criteria. In addition, the Ångström-exponent-based dust selection scheme may exclude some fine-mode dust particles, resulting in a potential deviation in the conversion factors. The two major gaps are the remote oceans and those polluted city regions (He et al., 2019b; Zhang et al., 2022). Ocean areas are less affected by complicated anthropogenic aerosols and may always be intruded by long-range transoceanic dust plumes

(Yu et al., 2021; Dai et al., 2022), which are thus preferentially focused on in this study.

To calculate the dust-related conversion factors at the oceanic/coast sites, we use a different scheme to identify the presence of dust in AERONET measurements. The new scheme is based on the particle linear depolarization ratio in the AERONET Version 3 aerosol inversion product, which is considered a good indicator for nonspherical dust particles (Shin et

al., 2018, 2019). It should be noted that the particle linear depolarization ratio values in AERONET retrieval are calculated from the combination of the particle size distribution and complex refractive index based on a spheroid light scattering model (Dubovik et al., 2006). Based on a modeling study, Gasteiger et al. (2011) found that the lidar-measured particle linear depolarization ratio values for pure mineral dust can be better reproduced by using an irregular particle shape assumption compared with using the spheroid shape assumption. Nevertheless, we consider it adequate to adopt AERONET-derived particle linear depolarization ratio values to qualitatively identify the presence of dust in the atmospheric column (Noh et al., 2017). Three factors motivate us to do so rather than to use the method raised by Ansmann et al. (2019b). First, applying AE <0.3 to select the dust-occurring data may exclude some data points representative of fine-mode dust particles that are proven to be present and cannot be ignored (Mamouri and Ansmann, 2014; Shin et al., 2019). Second, tracing the variations in dust-related conversion factors at different oceanic/coast sites may provide us with more information on the evolution of dust microphysical properties along with transoceanic transport routes (Rittmeister et al., 2017). In addition, marine aerosols that mainly consist of sea spray aerosols may occasionally show small AEs, which may confuse dust identification by using AE< 0.3 as a criterion (Smirnov et al., 2011; Yin et al., 2019).

In this study, we estimate the dust-related POLIPHON conversion factors at remote oceanic/coast sites using AERONET databases, which is considered an important step toward to the study of dust INPC/CCNC at a global scale and subsequent dust-induced ACI. The paper is organized as follows. We first briefly introduce the POLIPHON method and dust-related conversion factors as well as the AERONET data and dust data identification scheme. In section 3, we compare the dust-related conversion factor at the sites near deserts with the results from Ansmann et al. (2019b), represent the dust-related conversion factors at the ocean and coast sites, and discuss the possible reason behind the variation in conversion factors along the transoceanic transport. In the last section, summaries and conclusions are provided.

## 2 Data and methodology

### 2.1 POLIPHON method and conversion factors

The POLIPHON method can be used to obtain particle microphysical properties profiles (i.e., particle number, surface area, and volume concentration) and then, particle mass, CCN, and INP concentration profiles for several aerosol types using a combination of polarization lidar and sun photometer (Mamouri and Ansmann, 2014, 2015, 2016; Marinou et al., 2019). This method has been widely used in the estimations of CCN- and INP-relevant aerosol parameters for multiple aerosol types including dust (Ansmann et al., 2019a; Hofer et al., 2020; He et al., 2021b, 2022a, 2022b), marine aerosol (Mamouri and Ansmann, 2016, 2017; Haarig et al., 2019), continental aerosol (Mamouri and Ansmann, 2016, 2017), and smoke (Ansmann et al., 2021; Haarig et al., 2019). In this study, we focus on the dust-related CCN- and INP- properties.

The calculation is given in Table 1. First, the dust backscatter coefficient $\beta_d$ and extinction coefficient $\alpha_d$ can be derived by polarization lidar observations based on the inversion method from Fernald (1984) and the dust-component separation method from Tesche et al. (2009), as shown by the following equation:

$$\beta_{\mathrm{d}}(z) = \beta_{\mathrm{p}}(z) \frac{(\delta_{\mathrm{p}}(z) - \delta_{\mathrm{nd}})(1 + \delta_{\mathrm{d}})}{(\delta_{\mathrm{d}} - \delta_{\mathrm{nd}})(1 + \delta_{\mathrm{p}}(z))} \qquad (1)$$

where $\delta_{\mathrm{d}} = 0.31$ and $\delta_{\mathrm{nd}} = 0.05$ are dust (subscript 'd') and non-dust (subscript 'nd') particle depolarization ratio (Burton et al., 2013), respectively, and $\delta_{\mathrm{p}}$ and $\beta_{\mathrm{p}}$ are the lidar-derived particle depolarization ratio and backscatter coefficient, respectively. The lidar ratio for dust usually ranges from 30 sr to 60 sr depending on the different dust sources (Müller et al., 2007; Mamouri et al., 2013; Hu et al., 2020; Peng et al., 2021). Then, the dust extinction coefficient can be converted into particle mass concentration $M_{\mathrm{d}}$, particle number concentration $n_{250,\mathrm{d}}$ (r>250 nm), and particle surface concentration $s_{\mathrm{d}}$ and $s_{100,\mathrm{d}}$ (r>100 nm) by multiplying their corresponding conversion factors, i.e., $c_{\mathrm{v,d}}$, $c_{250,\mathrm{d}}$, $c_{\mathrm{s,d}}$, and $c_{\mathrm{s,100,d}}$. Finally, the dust-related INP concentration $n_{\mathrm{INP,d}}$ can be retrieved by inputting $n_{250,\mathrm{d}}$ and $s_{\mathrm{d}}$ into different INP parameterization schemes (DeMott et al., 2010, 2015; Niemand et al., 2012; Steinke et al. 2015; Ullrich et al., 2017). In addition, particle number concentration $n_{100,\mathrm{d}}$ (r>100 nm) is considered a good proxy for dust-related CCN concentration $n_{\mathrm{CCN,ss,d}}$ as discussed by Mamouri and Ansmann (2016). Here, the subscript 'ss' denotes the water supersaturation. Thus, $n_{\mathrm{CCN,ss,d}}$ can be obtained by multiplying $n_{100,\mathrm{d}}$ by a water supersaturation-dependent factor $f_{\mathrm{ss,d}}$, which is 1.0 for a typical liquid-water supersaturation value of 0.2%.

In the POLIPHON method, as introduced above, a series of conversion factors are essential to the conversion from the dust extinction coefficient to dust microphysical parameters regarding CCN and INP concentrations. The conversion factors are pre-calibrated from the historical database of sun photometer observations (denoted as j for each data point, counting from 1 to $J_{\mathrm{d}}$). The calculation processes are shown below (Ansmann et al., 2019b):

$$c_{\mathrm{v,d}} = \frac{1}{J_{\mathrm{d}}} \sum_{j=1}^{J_{\mathrm{d}}} \frac{V_{\mathrm{d},j}/D}{\tau_j/D} = \frac{1}{J_{\mathrm{d}}} \sum_{j=1}^{J_{\mathrm{d}}} \frac{v_{\mathrm{d},j}}{\alpha_{\mathrm{d},j}} \qquad (2)$$

$$c_{250,\mathrm{d}} = \frac{1}{J_{\mathrm{d}}} \sum_{j=1}^{J_{\mathrm{d}}} \frac{N_{250,\mathrm{d},j}/D}{\tau_j/D} = \frac{1}{J_{\mathrm{d}}} \sum_{j=1}^{J_{\mathrm{d}}} \frac{n_{250,\mathrm{d},j}}{\alpha_{\mathrm{d},j}} \qquad (3)$$

$$c_{\mathrm{s,d}} = \frac{1}{J_{\mathrm{d}}} \sum_{j=1}^{J_{\mathrm{d}}} \frac{S_{\mathrm{d},j}/D}{\tau_j/D} = \frac{1}{J_{\mathrm{d}}} \sum_{j=1}^{J_{\mathrm{d}}} \frac{s_{\mathrm{d},j}}{\alpha_{\mathrm{d},j}} \qquad (4)$$

$$c_{\mathrm{s,100,d}} = \frac{1}{J_{\mathrm{d}}} \sum_{j=1}^{J_{\mathrm{d}}} \frac{S_{100,\mathrm{d},j}/D}{\tau_j/D} = \frac{1}{J_{\mathrm{d}}} \sum_{j=1}^{J_{\mathrm{d}}} \frac{s_{100,\mathrm{d},j}}{\alpha_{\mathrm{d},j}} \qquad (5)$$

where $D$ is an introduced thickness for a given aerosol layer and $\tau$ is the AOD at 532 nm calculated from the sun-photometer-measured AOD at 500 nm together with the Ångström exponent (for 440-870 nm). $V_{\mathrm{d}}$ and $N_{250,\mathrm{d}}$ are the column particle volume concentration and column large particle (radius >250 nm) number concentration, respectively; $S_{\mathrm{d}}$ and $S_{100,\mathrm{d}}$ (radius >100 nm) are the column particle surface area concentrations. $V_{\mathrm{d}}$, $N_{250,\mathrm{d}}$, $S_{\mathrm{d}}$, and $S_{100,\mathrm{d}}$ are derived from the particle size distribution data in AERONET aerosol inversion products, which are introduced in detail in section 2.2. $\alpha_{\mathrm{d}}$ is the layer-

mean aerosol extinction coefficient at 532 nm; $v_d$ and $n_{250,d}$ are the layer-mean volume concentration and large particle (radius >250 nm) number concentration, respectively; and $s_d$ and $s_{100,d}$ (radius >100 nm) are the layer-mean particle surface area concentrations. The subscript 'd' denotes 'dust', which is considered the major contribution in column aerosol loading for the selected sun photometer data.

In addition, it is challenging to estimate the CCN concentration because the ability of aerosol particles to serve as CCN has a complex relationship with the particle hygroscopicity (associated with particle size and chemical composition) and water supersaturation level (Wang et al., 2010; Moore et al., 2012). As suggested by Shinozuka et al. (2015), a log-log regression analysis was performed to retrieve the CCN-relevant conversion factor $c_{100,d}$ and regression coefficient $\chi_d$ with the following equation:

$$\log(n_{100,d}) = \log(c_{100,d}) + \chi_d \log(\alpha_d) \tag{6}$$

In general, dust-related conversion factors have a regional-dependent characteristic, associated with the origin of dust regions as well as the specific local anthropogenic dust emissions (Philip et al., 2017). To extend the POLIPHON method toward global dust applications, Ansmann et al. (2019b) provided dust-related conversion factors at 20 AERONET sites in/near the typical desert regions, where other types of aerosols have less or even negligible contributions to the aerosol properties of the atmospheric column. Therefore, the obtained conversion factors can be considered representatives of pure or quasi-pure dust situations. To select the dust-occurring datasets for calculation, they adopted the constraints of column-integrated Ångström exponent (for 440–870 nm) <0.3 and aerosol optical depth (at 532nm) >0.1. Due to the influence of anthropogenic aerosols, the available dust-occurring data points are insufficient for calculating the conversion factors. To solve this issue, He et al. (2021b) applied simultaneous polarization lidar observations to assist in filtration of dust-occurring datasets in Wuhan (30.5°N, 114.4°E), a central Chinese megacity impacted by both local anthropogenic aerosol emissions and long-range transported dust (He and Yi, 2015; He et al., 2022c; Liu et al., 2022).

Dust plumes can also realize transoceanic transport in the Northern Hemisphere; there are two well-known pathways, i.e., the transatlantic route from the Saharan desert to America (Yu et al., 2021; Dai et al., 2022) and the transpacific route from Asian dust sources (Taklimakan Desert and Gobi) to America (Guo et al., 2017; Hu et al., 2019). However, dust-related conversion factors over the ocean are rarely reported. Sea spray aerosols usually have a large size, presenting a similar small Ångström exponent as dust particles (Haarig et al., 2017). To avoid the interference of sea spray aerosols, we use another scheme to identify the dust-occurring datasets, taking advantage of the particle linear depolarization ratio (PLDR) in AERONET aerosol inversion products (Shin et al. 2019). Therefore, sufficient available dust data points can be selected even at remote oceanic sites.

## 2.2 AERONET data and depolarization-ratio-based dust data point selection

AERONET is a global ground-based aerosol monitoring network and has provided more than 25 years of aerosol column property observations. CE-318 sun-sky photometers are used to measure direct solar irradiance (generally at 340, 380, 440,

500, 675, 870, 1020, and 1640 nm) and directional sky radiance to retrieve the spectral-resolved AODs and, in turn, the additional aerosol inversion products. Moreover, the latest applied CE318-T can also perform nighttime measurements of the spectral lunar irradiance. In this study, the AERONRT database (Holben et al., 1998; Dubovik et al., 2000; Dubovik and King, 2000) was employed to retrieve the dust-related conversion factors at 9 sites near/in deserts (to compare with the values given in Ansmann et al. (2019b) as a validation of the newly-proposed dust selection scheme) and at 20 oceanic/coast sites influenced by long-range transported dust (see figure 1). The oceanic/coast sites can be classified into five regional clusters, i.e., Pacific, Pacific coast, Atlantic, Indian Ocean, and Arctic Ocean, representing dust characteristics in different regions.

We used the quality-assured level-2.0 AOD in AERONET Version 3 aerosol optical depth-solar products (Giles et al., 2019). Moreover, the particle volume size distribution with 22 size bins (radius) ranging from 50 nm to 15 μm and PLDR in AERONET Version 3 aerosol inversion products are also used for calculating $V_d$, $N_{250,d}$, $S_d$, and $S_{100,d}$ as described in section 2.1 (Sinyuk et al., 2020). The specific calculation processes can be found in Mamouri and Ansmann (2014,2015) and Ansmann et al. (2019b). For the near/in desert sites, level-2.0 (quality-assured) aerosol inversion products are applied, while for the oceanic/coast sites, level-1.5 (cloud-screened and quality-controlled) aerosol inversion products are applied since the level-2.0 PLDR data are unavailable. The basic information of the selected sites is shown in Table 2, including period, longitude, latitude, and the number of data points for total, dust-dominated mixture, and pure dust.

In AERONET retrieval, based on the aerosol spheroid model, the combination of the particle size distribution and complex refractive index can be used to further compute the two elements of the Müller scattering matrix, i.e., $F_{11}(\lambda)$ and $F_{22}(\lambda)$ (Bohren and Huffman, 1983; Dubovik et al., 2006; Shin et al., 2018), which can then be used to derive the backscattering PLDR with the following formula:

$$\delta_\lambda^p = \frac{1 - F_{22}(\lambda, 180°)/F_{11}(\lambda, 180°)}{1 + F_{22}(\lambda, 180°)/F_{11}(\lambda, 180°)} \tag{7}$$

AERONET PLDR data are a good indicator of dust occurrence and have been verified to be well correlated with lidar-derived values (Noh et al., 2017). Shin et al. (2018) found that PLDR values at 870 and 1020 nm are more reliable according to the comparison with those from lidar observations for pure dust particles. Therefore, we use PLDR at 1020 nm $\delta_{1020}^p$ (only denote as 'PLDR' hereafter) to select the dust-occurring data points for POLIPHON conversion factor calculation (Shin et al., 2019). Note that the overestimation of near-infrared PLDR is reported by comparison with concurrent polarization lidar observations (Toledano et al., 2019; Haarig et al., 2022), possibly due to the assumption of the spheroid particle in AERONET inversion. Nevertheless, $\delta_{1020}^p$ values are only used to qualitatively identify the dust presence with the presupposed threshold values. Its validity will be verified by comparing the derived conversion factors with those from Ansmann et al. (2019b) in Section 3.1.

The column-integrated dust ratio ($R_{d,1020}$), representing the contribution proportion of dust backscatter to the total particle backscatter in the atmospheric column, is defined as follows:

$$R_{d,1020} = \frac{\left(\delta^p_{1020} - \delta^p_{nd}\right)\left(1 + \delta^p_d\right)}{\left(\delta^p_d - \delta^p_{nd}\right)\left(1 + \delta^p_{1020}\right)} \tag{8}$$

It should be noted that the dust ($\delta^p_d$) and non-dust ($\delta^p_{nd}$) particle depolarization ratios are set to 0.30 and 0.02, respectively, to be consistent with the value proposed by Shin et al. (2019). These two values are slightly different from those used in the POLIPHON method in Eq. (1). According to the scheme from Shin et al. (2019), we classify data points with $R_{d,1020}$ values of >0.89 as pure dust (PD) and 0.53-0.89 as dust-dominated mixture (DDM, which can also be considered 'mixed dust', as in He et al. (2019b). Note that in this study, DDM includes the combination of sectors B (fine-mode fraction (FMF) =0–0.4), C (FMF= 0.4–0.6), and E (FMF= 0.6–1.0) in Shin et al. (2019). A flow chart for dust-occurring data point selection and dust-related conversion factor retrieval is shown in figure 2. For the near/in desert sites, we presented the results from the PD cluster, which is adequate for calculating the conversion factors. For the oceanic/coast sites, the results from both the PD and PD+DDM clusters are provided for comparison.

## 3 Dust-related conversion factors from the AERONET database

In this section, we mainly focus on the calculation of the dust-related conversion factors in the POLOPHON method with the new dust identification scheme, which is based on the particle linear depolarization ratio in the AERONET data product. To verify the performance of the proposed dust identification scheme, the dust-related conversion factors near deserts are first calculated at nine AERONET sites and compared with those obtained by Ansmann et al. (2019b). Then, the dust-related conversion factors $c_{v,d}$, $c_{250,d}$, $c_{s,d}$, $c_{s,100,d}$, $c_{100,d}$, and $\chi_d$ at 20 oceanic/coastal AERONET sites are derived with the proposed method. Finally, the variations in the dust-related conversion factors along the two transoceanic (i.e., transatlantic and transpacific) pathways are analyzed.

### 3.1 Intercomparison of dust-related conversion factors near deserts with retrievals in Ansmann et al. (2019b)

To validate the performance of the newly-proposed dust dataset selection scheme, we chose 9 out of 20 AERONET sites used in Ansmann et al. (2019b) to compare the obtained conversion factors. They are mainly from three typical regions, including North Africa, the Middle East, and Asia (see figure 1). It should be mentioned that $c_{100,d}$ values tend to be divergent when the aerosol extinction coefficient is larger than 600 Mm$^{-1}$. Hence, the validations were only performed for $c_{v,d}$, $c_{250,d}$, $c_{s,d}$, and $c_{s,100,d}$. Figure 3 shows the scatters regarding the relationships between 532-nm aerosol extinction and $n_{250,d}$, $v_d$, $s_d$, and $s_{100,d}$ for pure data situations at Cape Verde, Dushanbe, and Mezaira'a. The conversion factor values are also given accordingly. Here, level 2.0 AOD and aerosol inversion product data were employed. PD datasets selected with the method from Shin et al. (2019) perform well, as a highly linear correlation can be found, with linear Pearson correlation coefficients exceeding 0.9.

In addition to the three sites in figure 3, the conversion factors for the remaining six sites are shown in Table 3 as well. The results for both PD and PD+DDM clusters are provided here. Generally, regional differences in dust characteristics can be found in different dust sources. $c_{v,d}$, $c_{250,d}$, and $c_{s,d}$ calculated from the PD and PD+DDM datasets are almost consistent with each other, suggesting the absolute dominance of dust particles at these near/in desert sites. Interestingly, there are larger differences in $c_{s,d}$ and $c_{s,100,d}$ between the PD and DDM clusters at two Middle East sites, i.e., Eilat and Mezaira'a. After a careful check, it is noted that the DDM datasets have aerosol extinction coefficient values of 300-600 Mm$^{-1}$ and show significantly larger $s_d$ and $s_{100,d}$ than those for PD datasets (see figure A1). The special pattern reflects the involvement of a specific type of local aerosol in the dust-dominated mixture. Moreover, the PD-DDM differences are even larger for $c_{s,100,d}$ at both sites, indicating that the additional involved aerosols may play a vital role at the particle size spectral of >100 nm. According to further examination, this special pattern can generally be found at partial sites from the Middle East, Africa, and polluted European cities, which, however, are rarely present at sites from East Asia, Australia, South America, and North America. Thus, it should be noted that more care should be taken when employing DDM data to retrieve dust-related conversion factors at terrestrial sites in the Middle East, Africa, and polluted European cities in future work.

For comparison, we also plot the conversion factors in figure 4 together with those given by Ansmann et al. (2019b) (hereafter denoted as 'A-19'). $c_{v,d}$ and $c_{250,d}$ calculated from three dust datasets (PD, DDM+PD, and A-19) coincide with each other very well. The relative differences between either A-19 and PD or A-19 and DDM+PD are generally as small as <8.5% for $c_{250,d}$ (except for 19.2% at the DA site) and <12.5% for $c_{v,d}$ (except for 20.0% at the DA site). Compared with A-19, $c_{s,d}$ values calculated from PD datasets generally show relative differences of <16.5% (except for 23.2% at the CV site); in contrast, $c_{s,d}$ differences between DDM+PD datasets and A-19 are much larger (up to <36.2%). For $c_{s,100,d}$, compared with the results from A-19, values calculated from PD datasets show relative differences as high as <36.2%, while values from PD+DDM datasets show relatively larger relative differences up to <42.5%. Using either the FMF (Lee et al., 2010) or Ångström exponent (Ansmann et al., 2019b) in the AERONET data as the dust criterion implies an assumption that dust is constrained to the coarse mode. However, the proportion between fine- and coarse- mode dust may be altered during transport due to the quicker removal of dust particles with larger sizes (Yu et al., 2021); hence, the dust criterion AE <0.3 may exclude a portion of fine-mode dust (with radius <100 nm) dominated cases, resulting in a general underestimation of $c_{s,d}$. Furthermore, region-featured emissions of non-dust small particles are also possibly responsible for this discrepancy.

**3.2 Dust-related conversion factors $c_{v,d}$, $c_{250,d}$, $c_{s,d}$, and $c_{s,100,d}$ at the ocean and coast sites**

In this study, we chose 20 ocean and coast AERONET sites, which were classified into five region categories including the Pacific, Pacific coast (both east and west coasts), Atlantic, Indian Ocean, and Arctic Ocean (Huang et al., 2015; Zhao et al., 2022). Figure 5 shows the relationships between 532-nm aerosol extinction and $n_{250,d}$, $v_d$, $s_d$, and $s_{100,d}$ for both the PD and PD+DDM cases at Mauna Loa (middle Pacific), Shirahama (west Pacific coast), Tudor Hill (west Atlantic), and Amsterdam Island (south Indian Ocean). The conversion factor values are also given accordingly. To eliminate abnormal

values and retain the available dust data as much as possible, the data points with aerosol extinction of >20 Mm$^{-1}$ are considered in the calculation. PD data points generally show a good linear correlation. For the PD+DDM cluster, three island sites show a similar good correlation except for Shirahama, which is due to the small contribution of marine aerosols to total column aerosol loading (usually with a global mean AOD of ~0.05, Smirnov et al., 2009). At the Shirahama site, it is conjectured that anthropogenic aerosols considerably contribute to the column aerosol loading and lead to the spread of scatters, which reflect variations in the characteristics (size distribution, complex refractive index, and so on) of other aerosol components in the dust-dominated mixture.

The conversion factors $c_{v,d}$, $c_{250,d}$, $c_{s,d}$, and $c_{s,100,d}$ at the other oceanic/coast sites are also provided in Table 4. The results for only the PD cluster and combined PD and DDM clusters are listed. We consider the conversion factors with ≥12 available PD data points valid (provided in Table 4). Moreover, to guarantee robustness, only the retrieved conversion factors with the linear Pearson correlation coefficient $R$ exceeding 0.70 are considered valid, except for PD-derived $c_{250,d}$ values at NR (R=0.32) and AS (R=0.50), which should especially be handled with care in scientific applications. We also plot these dust-related conversion factors in figure 6 for comparison. According to the estimation in section 3.1, it is suggested to preferentially use the PD datasets in the calculation to avoid the potential contribution of specific local aerosols, e.g., the special pattern in $c_{s,d}$ and $c_{s,100,d}$ at Eilat and Mezaira'a (see section 3.1). Nevertheless, PD+DDM datasets may take part in the calculation as a suboptimal option if the sole use of PD datasets cannot guarantee the validity of conversion factors (10 out of 20 sites).

As seen in figure 6, $c_{250,d}$ mainly ranges from 0.17 to 0.28 Mm cm$^{-3}$; the $c_{250,d}$ values calculated from the PD and PD+DDM datasets agree with each other very well except for the Midway Island and Nauru sites, indicating that few non-dust aerosols contribute to the particle size spectra of >250 nm and that $c_{250,d}$ is a relatively stable factor from region to region. For $c_{s,d}$ and $c_{s,100,d}$, their values from the DDM+PD datasets mainly have a systematically positive deviation (<25%), compared with those from the PD datasets, which may be affected by marine aerosols. In addition, $c_{s,d}$ and $c_{s,100,d}$ values at coastal sites (especially at Shirahama, Osaka, and Hokkaido University) are considerably larger than those at remote ocean sites, revealing their higher sensitivity (compared with $c_{250,d}$) to the involvement of other aerosols. However, large differences in $c_{v,d}$ values are found not only between the PD and PD+DDM datasets but also from region to region. He et al. (2019b) also reported that mixed dust in Wuhan has a smaller $c_{v,d}$ than pure dust near the source region of Asian dust. Another interesting finding for $c_{v,d}$ is that the values in the Arctic are only half of those in other regions. PD data points are rarely identified in the Arctic and conversion factors are all calculated from the DDM datasets here; abundant other aerosol types in the Arctic, e.g., smoke, anthropogenic aerosol, and marine aerosol (Engelmann et al., 2021; Zhao et al., 2022), may account for the low $c_{v,d}$ values. Nevertheless, $c_{v,d}$ can be beneficial to the validation of the mass extinction efficiency, a variable combining $c_{v,d}$ and dust density, in the 3-D global dust model (Adebiyi et al., 2020; Kok et al., 2021b; Wang et al., 2021; Wang et al., 2022).

The region-to-region variations in the conversion factors (i.e., $c_{v,d}$, $c_{s,d}$, and $c_{s,100,d}$) can be clearly found, as shown in figure 6. Although it is difficult to quantitatively study the reasons behind this region-dependent feature, one can first

attribute this to the diverse contributions from different dust sources. Excluding some occasional extreme events (Uno et al., 2009), a given oceanic region is generally influenced by specific dust sources via typical dust transport pathways. In the middle- and low-latitude Atlantic, the primary dust transport pathway is from the Saharan desert in North Africa to the eastern coastal regions of North America (Rittmeister et al, 2017; Yu et al., 2021). In the North Atlantic, Baddock et al. (2017) reported that dust aerosols are mainly from Iceland. Dust aerosols in the Arctic are more complicated, coming from

high-latitude dust sources in the Northern Hemisphere (e.g., Alaska, Canada, Northern Europe, and Russia) (Bullard et al., 2016; Meinander et al., 2022), Arctic local sources (Shi et al., 2022), Asia (Zhao et al., 2022), and North Africa (Shi et al., 2022). For the Pacific, dust aerosols mainly originate from the Central and East Asian dust sources and transport to North America (Guo et al., 2017; Hu et al., 2019). At the remaining oceanic sites in the Southern Hemisphere, dust aerosols can be related to Australia, New Zealand, Patagonia, and Southern Africa (Bullard et al., 2016; Struve et al., 2020; Kok et al., 2021a;

Meinander et al., 2022). In addition, as the downstream areas, the possible aging and mixing of dust with other aerosol types during long-range transport may also be responsible for the region-to-region variations in conversion factors (Kim and Park, 2012; Goel et al., 2020).

### 3.3 Dust-related conversion factors $c_{100,d}$ and $\chi_d$ at the ocean and coast sites

In addition to the INP-relevant conversion factors, the relationship between 532-nm aerosol extinction and CCN-relevant

parameters $n_{100,d}$ is also studied in this section. The analysis is based on the relationship between $\log(n_{100,d})$ and $\log(\alpha_d)$ as reported by Shinozuka et al. (2015). Figure 7 shows the relationship between the aerosol extinction and particle number concentration (radius >100 nm) $n_{100,d}$ at nine oceanic/coast sites. The data points representing PD (in blue) and DDM+PD (in orange) are both plotted. Ansmann et al. (2019b) found that $\log(n_{100,d})$ and $\log(\alpha_d)$ are strongly correlated when taking data points with $\alpha_d$ in the range of 100-600 Mm$^{-1}$ into consideration, while the correlation strength significantly decreases

and the data points trend to be dispersive once $\alpha_d$ values exceed 600 Mm$^{-1}$. The AERONET sites selected here generally show a clear atmospheric environment with limited pollution aerosols and can generally fulfill the constraint of $\alpha_d < 600$ Mm$^{-1}$ except for those coast sites, e.g., Shirahama and Osaka. To retain sufficient data points, we adopted data points with $\alpha_d$ values ranging from 20 to 600 Mm$^{-1}$ in our calculation.

Table 5 lists the values of $c_{100,d}$ and $\chi_d$ for both the PD and DDM+PD datasets. Considering the regression coefficient

$\chi_d > 0.50$ as valid analysis, attention should be given when using the results at the coast sites Osaka, American Samoa, and Amsterdam Island. The PD and DDM+PD datasets generally show a similar slope (corresponding to $\chi_d$) in regression analysis, except for the Osaka site, for which an evident intersection between two fitted lines appears, attributed to the sparse PD data points available for fitting. Moreover, it should be mentioned that using the newly-proposed dust dataset selection

scheme to retrieve the CCN-relevant conversion factors seems not robust on the continent. Thus, more care should be taken

when retrieving $c_{100,d}$ and $\chi_d$ for those polluted city regions in future work.

**3.4 Variations in conversion factors along dust transport paths**

The dust-related conversion factors may significantly vary along the way of dust transport due to the potential modifications of dust microphysical properties caused by particle sedimentation, aging processes, external mixing with other aerosols, and so on. There are two main transoceanic paths of dust transport, i.e., the transatlantic path from the Saharan

Desert to America (Rittmeister et al, 2017; Yu et al., 2021; Dai et al., 2022) and the transpacific path from Asian dust sources (Taklimakan Desert and Gobi) to America (Guo et al., 2017; Hu et al., 2019). Here we selected several sites along these two paths to evaluate the variations in conversion factors. For the transpacific transport, six sites from Asian dust sources to America were selected, including Dushanbe, SACOL in Lanzhou, Shirahama, Midway Island, Mauna Loa, and Trinidad Head. For the Transatlantic transport, four sites from North Africa to America were selected, including Dakar, Cape

Verde, ARM Graciosa, and Tudor Hill.

Figure 8 shows the conversion factors $c_{v,d}$, $c_{250,d}$, $c_{s,d}$, and $c_{s,100,d}$ at the selected sites along the two dust transport paths. These conversion factors are calculated from the PD+DDM cluster. It is noted that the microphysical properties of dust particles originating from the Saharan Desert and Asian dust sources are very different. Moreover, with the increase in transport distances, an evident variation tendency is observed for all the conversion factors at both transoceanic paths. $c_{v,d}$

values show a significant decline along with transport, indicating that the proportion of dust particles in the atmospheric column tends to be smaller due to sedimentation. He et al. (2021b) also observed a relatively smaller $c_{v,d}$ of $0.52\times10^{-12}$ Mm m$^3$ m$^{-3}$ in the downstream area in central China compared with the value obtained near the sources of Asian dust. $c_{250,d}$ values show a gradual increase trend along with transport, suggesting the increased contribution of large-size sea spray aerosols in the atmospheric column. For $c_{s,d}$, a general decline is observed between the values before and after transoceanic

transport. A plunge of $c_{s,d}$ is prominent for the transpacific path. In contrast, $c_{s,100,d}$ values only show an apparent enhancement for the transatlantic path, while the variation trend for the transpacific path is generally inapparent. This suggests that particles with radii <100 nm should be responsible for the decrease in $c_{s,d}$ after transoceanic transport.

**4 Summary and conclusions**

To improve the current consideration of ACIs in atmospheric circulation models, it is necessary to characterize the 3-D

distribution of dust-related CCNC and INPC at a global scale. The combination of CALIOP spaceborne lidar observations and the POLIPHON method has the potential to realize this purpose. In this study, as the first step, we retrieved the essential dust-related conversion factors at remote ocean sites where these parameters are less constrained. Historical AERONET databases were employed to calculate the conversion factors. Depolarization ratios at 1020 nm from the AERONET version

3 aerosol inversion product were used to calculate the column-integrated dust ratios $R_{d,1020}$, which were further applied to identify the dust presence within the atmospheric column (Shin et al., 2018, 2019). Compared with the use of the Ångström exponent (Ansmann et al., 2019b), this treatment is beneficial for containing fine-mode dust dominated cases (after the preferential removal of large-size dust particles during transport), mitigating the occasional interference of large-size marine aerosols, and studying the evolution of dust microphysical properties along the transoceanic transport path.

It is found that $c_{v,d}$, $c_{250,d}$, and $c_{s,100,d}$ are generally consistent with those provided by Ansmann et al. (2019b) at nine sites near deserts. However, the $c_{s,d}$ values obtained in this study are systematically larger than those given by Ansmann et al. (2019b), which is attributed to the possible miss of fine-mode dust particles with radii <100 nm. For all the dust-related conversion factors, the PD and PD+DDM datasets give similar results except for two Middle East sites, i.e., Eilat and Merzaira'a. Then, we calculated all the dust-related conversion factors at 20 oceanic/coast sites using both the PD and PD+DDM datasets. Only 10 sites have adequate PD data points to retrieve $c_{v,d}$, $c_{250,d}$, $c_{s,d}$, and $c_{s,100,d}$. Among them, $c_{v,d}$ values are more sensitive to the influence of other aerosols involved in the atmospheric column and show large differences between the PD and PD+DDM clusters as well as from region to region. In addition, only 9 sites successfully obtained the CCN-relevant factors $c_{100,d}$ and $\chi_d$ in the regression analysis. In addition, $c_{v,d}$ values gradually decrease along with transoceanic transport; in contrast, $c_{250,d}$ values show an increasing trend. A general decline in $c_{s,d}$ can be found after transoceanic transport; however, this decrease is not observed for $c_{s,100,d}$, suggesting that the discrepancy may be due to the influence of the particle size spectral of <100 nm (radius).

For ocean sites, the depolarization-ratio-based method for selecting dust-occurring data is proven to be valid and feasible. The PD datasets are suggested to be the preferential option to calculate the dust-related conversion factors. If the available PD data points are insufficient, the PD+DDM cluster would be a suboptimal option allowing us to obtain the conversion factors with certain accuracy and robustness. In future work, we will conduct case studies on dust-cloud interactions over the ocean with CALIOP spaceborne lidar observations and the dust-related conversion factors used in this study. In addition, the dust-related conversion factors at polluted city sites will be examined with the same method; under this situation, the application of PD or PD+DDM datasets needs to be further discussed in depth. Once those conversion factors at polluted city sites are retrieved, a global dust-related conversion factor grid dataset will possibly be obtained by geographical interpolation. After that, the 3-D view of global CCNC and INPC can be anticipated to improve our current consideration of ACIs in atmospheric circulation models.

**Appendix A: Additional analysis**

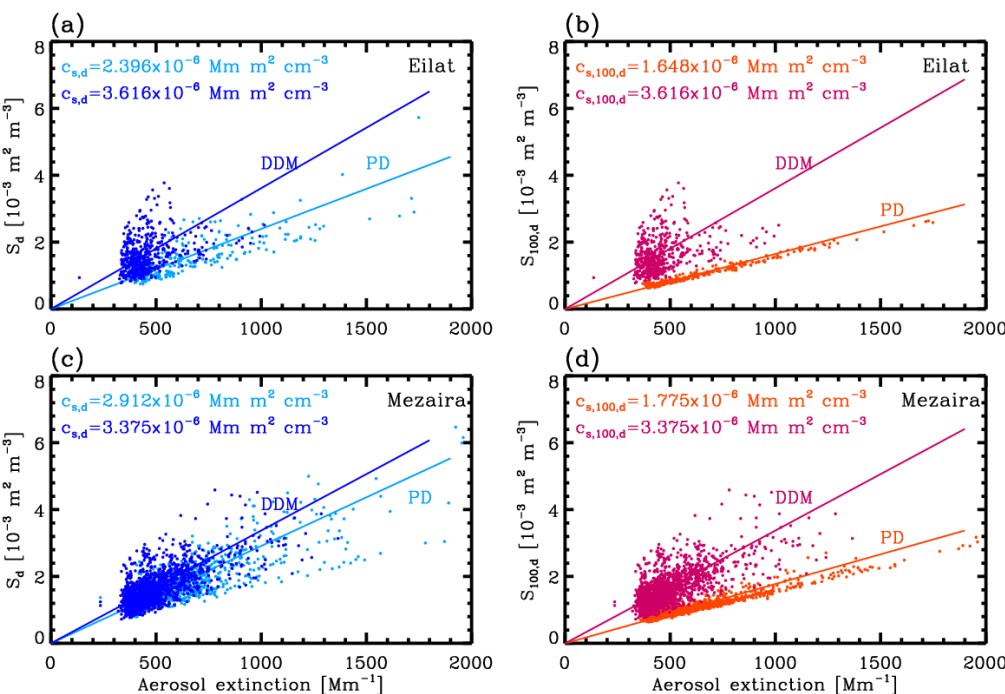

**Figure A1: Relationship between aerosol extinction coefficient at 532 nm and surface area concentration $s_d$ and $s_{100,d}$ (only considering particles with radius >100 nm) for pure dust (PD) and dust-dominated mixture (DDM) at two Middle East sites, i.e., (a) and (b) for Eilat, and (c) and (d) for Merzaira'a. The PD and DDM data points are determined by the AERONET V3 database (level-2.0 AOD products and level-2.0 aerosol inversions) according to the method from Shin et al. (2019). The corresponding dust-related conversion factors $c_{s,d}$ and $c_{s,100,d}$ are also given, respectively.**

**Data availability**

All data used in this work can be accessed through the AERONET home page at https://aeronet.gsfc.nasa.gov/ (AERONET, 2022).

**Author Contributions**

Yun He conceived the research, analyzed the data, acquired the research funding, and wrote the manuscript. Zhenping Yin conceived the research, participated in scientific discussions, and reviewed and proofread the manuscript. Albert Ansmann

reviewed the manuscript and participated in scientific discussions. Fuchao Liu and Longlong Wang reviewed and proofread the manuscript. Dongzhe Jing and Huijia Shen participated in the data processing.

**Competing interests**

The authors declare that they have no conflict of interest.

**Acknowledgments**

The authors thank all PIs of the AERONET sites used in this study for maintaining their instruments and providing their data to the community.

**Financial support**

This research has been supported by the National Natural Science Foundation of China (grant nos. 42005101, 41927804, and 42205130), the Fundamental Research Funds for the Central Universities (grant no. 2042021kf1066), the Natural Science

Foundation of Hubei Province (grant no. 2021CFB406), the Innovation and Development Project of China Meteorological Administration (grant no. CXFZ2022J060), the Chinese Scholarship Council (CSC) (grant no. 202206275006), and the Meridian Space Weather Monitoring Project (China).

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

**Table 1. Dust-related parameters of optical properties, cloud condensation nuclei, and ice-nucleating particles calculated by the POLIPHON method (Tesche et al., 2009; Marinou et al., 2019; Ansmann et al., 2019b). D10, D15, U17, N12, and S15 refer to the respective INP parameterization in DeMott et al. (2010), DeMott et al. (2015), Ullrich et al. (2017), Niemand et al. (2012), and Steinke et al. (2015). The subscript 'ss' denotes water supersaturation. The uncertainties are provided based on using the CALIOP level-2 aerosol profile product for INPC and CCNC calculation.**

| Dust-related parameters | Computation | Uncertainty |
|---|---|---|
| Dust backscatter $\beta_{\mathrm{d}}$ (Mm$^{-1}$ sr$^{-1}$) | $$\beta_{\mathrm{d}}(z) = \beta_{\mathrm{p}}(z)\frac{(\delta_{\mathrm{p}}(z) - \delta_{\mathrm{nd}})(1 + \delta_{\mathrm{d}})}{(\delta_{\mathrm{d}} - \delta_{\mathrm{nd}})\left(1 + \delta_{\mathrm{p}}(z)\right)}$$ $$\delta_{\mathrm{d}} = 0.31,\ \delta_{\mathrm{nd}} = 0.05$$ | <49% |
| Dust extinction $\alpha_{\mathrm{d}}$ (Mm$^{-1}$) | $$\alpha_{\mathrm{d}}(z) = LR_{\mathrm{d}} \times \beta_{\mathrm{d}}(z)$$ | <59% |
| Dust mass conc. $M_{\mathrm{d}}$ (µg m$^{-3}$) | $$M_{\mathrm{d}}(z) = \rho_{\mathrm{d}} \times \alpha_{\mathrm{d}}(z) \times c_{\mathrm{v,d}}$$ $$\rho_{\mathrm{d}} = 2.6\ \mathrm{g\ cm}^{-3}$$ | <60% |
| Particle number conc. (r> 250 nm) $n_{250,\mathrm{d}}$ (cm$^{-3}$) | $$n_{250,\mathrm{d}}(z) = \alpha_{\mathrm{d}}(z) \times c_{250,\mathrm{d}}$$ | <60% |
| Particle surface conc. $S_{\mathrm{d}}$ (m$^2$ cm$^{-3}$) | $$S_{\mathrm{d}}(z) = \alpha_{\mathrm{d}}(z) \times c_{\mathrm{s,d}}$$ | <60% |
| Particle surface conc. (r> 100 nm) $S_{100,\mathrm{d}}$ (m$^2$ cm$^{-3}$) | $$S_{100,\mathrm{d}}(z) = \alpha_{\mathrm{d}}(z) \times c_{\mathrm{s,100,d}}$$ | <60% |
| $n_{\mathrm{INP,d}}$ (L$^{-1}$) from $n_{250,\mathrm{d}}$ | INP parameterization D10 and D15 | <500% |
| $n_{\mathrm{INP,d}}$ (L$^{-1}$) from $S_{\mathrm{d}}$ and $S_{100,\mathrm{d}}$ | INP parameterization U17, N12 and S15 | <500% |
| $n_{\mathrm{CCN,ss,d}}$ (L$^{-1}$) from $n_{100,\mathrm{d}}$ | $$n_{100,\mathrm{d}}(z) = c_{100,\mathrm{d}} \times \alpha_{\mathrm{d}}(z)^{\chi_d}$$ $$n_{\mathrm{CCN,ss,d}}(z) = f_{\mathrm{ss,d}} \times n_{100,\mathrm{d}}(z)$$ | <200% |


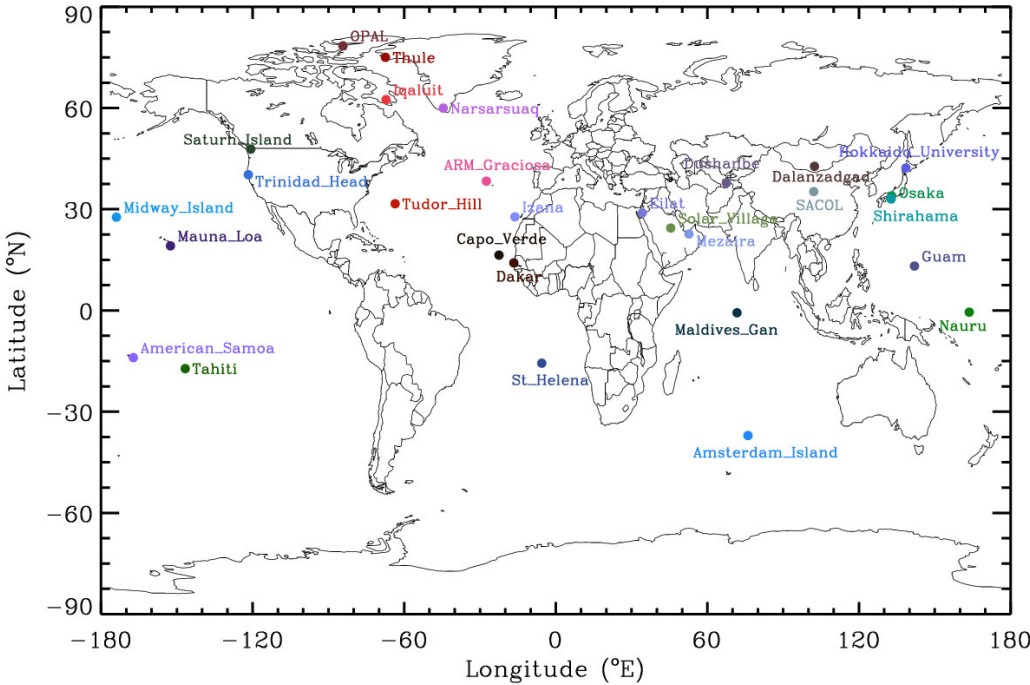

**Figure 1: Overview of the oceanic/coast (20 sites) and near-desert (9 sites selected from Ansmann et al. (2019b)) AERONET sites used in this study. Labels for each site are taken from AERONET site list.**


**Table 2. Basic information of the AERONET sites selected for dust-related POLIPHON conversion factors calculation near the deserts (from North Africa, the Middle East, and Asia) and over the ocean and coast (from the Pacific, Pacific coast, Atlantic, Indian Ocean, and Arctic Ocean) regions, including the site name and abbreviation, period, and location. The available number of data points for total, dust-dominated mixture (DDM), and pure dust (PD) in AERONET version 3 aerosol inversion products (level 2.0 for near-desert cluster and level 1.5 for oceanic/coast cluster) are also given, respectively. Average AODs at 532 nm (AOD) and Ångström exponents between 440 nm and 870 nm (AE) at each site are provided, respectively.**


|  | Site (Abbreviation) | Period | Location | Total Obs. | DDM Obs. | PD Obs. | Total AOD | Total AE |
|---|---|---|---|---|---|---|---|---|
| Pacific | Tahiti (TA) | 1999-2010 | 41.05N, 124.15W | 1210 | 247 | 6 | 0.06 | 0.62 |
|  | Nauru (NR) | 1999-2013 | 0.52S, 166.92E | 1379 | 141 | 12 | 0.06 | 0.48 |
|  | Midway_Island (MI) | 2001-2015 | 28.21N, 177.38W | 1663 | 351 | 26 | 0.09 | 0.56 |
|  | American_Samoa (AS) | 2014-2022 | 14.25S, 170.56W | 1080 | 197 | 14 | 0.07 | 0.49 |
|  | Guam (GA) | 2006-2009 | 13.43N, 144.80E | 239 | 33 | 2 | 0.08 | 0.49 |
|  | Mauna_Loa (ML) | 1994-2022 | 19.54N, 155.58W | 30384 | 1481 | 27 | 0.02 | 1.14 |
| Pacific Coast | Hokkaido_University (HU) | 2015-2022 | 43.08N, 141.34E | 3524 | 321 | 11 | 0.18 | 1.36 |
|  | Osaka (OS) | 2000-2022 | 34.65N, 135.59E | 3296 | 99 | 13 | 0.23 | 1.33 |
|  | Shirahama (SH) | 2000-2022 | 33.69N, 135.36E | 9095 | 1421 | 68 | 0.23 | 1.24 |
|  | Saturn_Island (SI) | 1997-2022 | 48.78N, 123.13W | 5986 | 527 | 0 | 0.12 | 1.35 |
|  | Trinidad_Head (TR) | 2005-2017 | 41.05N, 124.15W | 3978 | 678 | 18 | 0.09 | 0.88 |
| Atlantic | ARM_Graciosa (AG) | 2013-2022 | 39.09N, 28.03W | 2549 | 218 | 50 | 0.09 | 0.68 |
|  | Tudor_Hill (TH) | 2007-2022 | 32.26N, 64.88W | 2222 | 502 | 40 | 0.10 | 0.84 |
|  | St_Helena (ST) | 2016-2022 | 15.94S, 5.67W | 294 | 18 | 0 | 0.07 | 0.93 |
| Indian Ocean | Maldives_Gan (MG) | 2018-2022 | 0.69S, 73.15E | 1153 | 190 | 4 | 0.11 | 0.80 |
|  | Amsterdam_Island (AI) | 2002-2022 | 37.80S, 77.57E | 1241 | 216 | 25 | 0.07 | 0.43 |
| Arctic Ocean | Narsarsuaq (NA) | 2013-2022 | 61.16N, 45.42W | 2918 | 139 | 2 | 0.06 | 1.35 |
|  | Thule (TL) | 2007-2022 | 76.52N, 68.77W | 4604 | 119 | 0 | 0.06 | 1.44 |
|  | OPAL (OP) | 2007-2022 | 79.99N, 85.94W | 2223 | 34 | 0 | 0.07 | 1.52 |
|  | Iqaluit (IQ) | 2008-2020 | 63.75N, 68.54W | 1190 | 60 | 0 | 0.08 | 1.54 |
| North Africa | Cape_Verde (CV) | 1994-2022 | 16.73N, 22.94W | 6020 | 174 | 1912 | 0.35 | 0.29 |
|  | Dakar (DK) | 1996-2020 | 14.39N, 16.96W | 9674 | 975 | 3620 | 0.40 | 0.35 |
|  | Izana (IZ) | 1997-2022 | 28.31N, 16.50W | 5114 | 0 | 87 | 0.05 | 0.92 |
| Middle East | Eilat (EI) | 2007-2022 | 29.50N, 34.92E | 9290 | 503 | 263 | 0.19 | 0.83 |
|  | Solar_Village (SV) | 1999-2015 | 24.907N, 46.40E | 14278 | 1839 | 2234 | 0.32 | 0.53 |
|  | Mezaira'a (ME) | 2004-2022 | 23.11N, 53.76E | 8679 | 1672 | 998 | 0.34 | 0.66 |
| Asia | Dushanbe (DU) | 2010-2022 | 38.55N, 68.86E | 4939 | 621 | 331 | 0.26 | 0.78 |

| | Dalanzadgad (DA) | 1997-2022 | 43.58N,104.42E | 3864 | 10 | 12 | 0.10 | 1.06 |
|---|---|---|---|---|---|---|---|---|
| | SACOL(LA) | 2006-2013 | 35.95N,104.14E | 3382 | 317 | 186 | 0.32 | 0.93 |

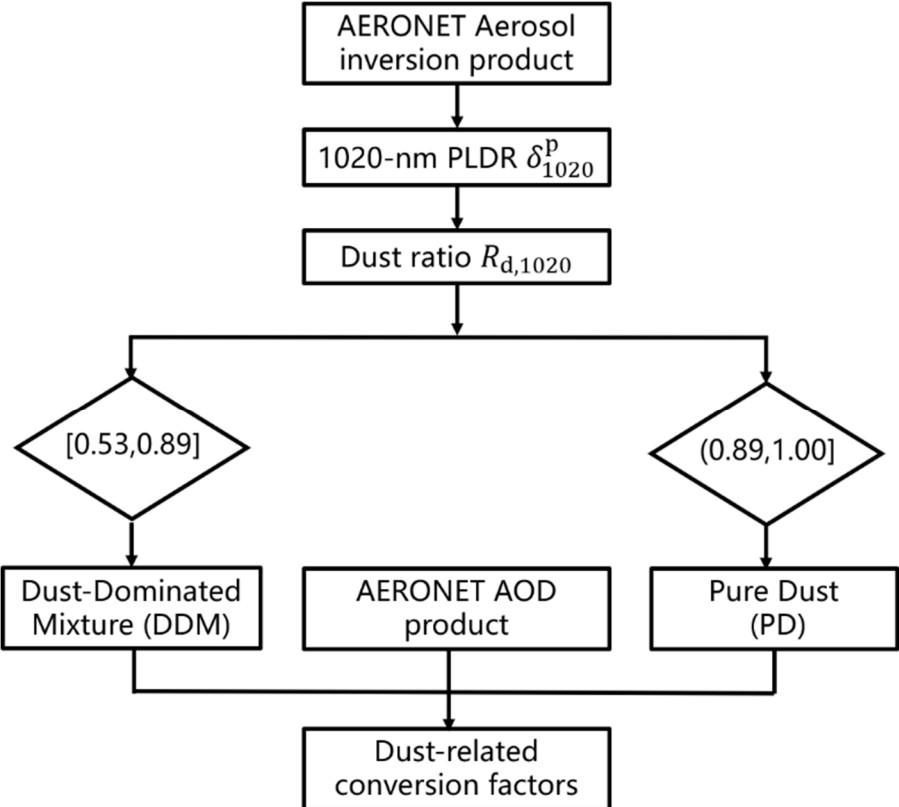

**Figure 2: Flow chart for retrieving the dust-related POLIPHON conversion factors.**

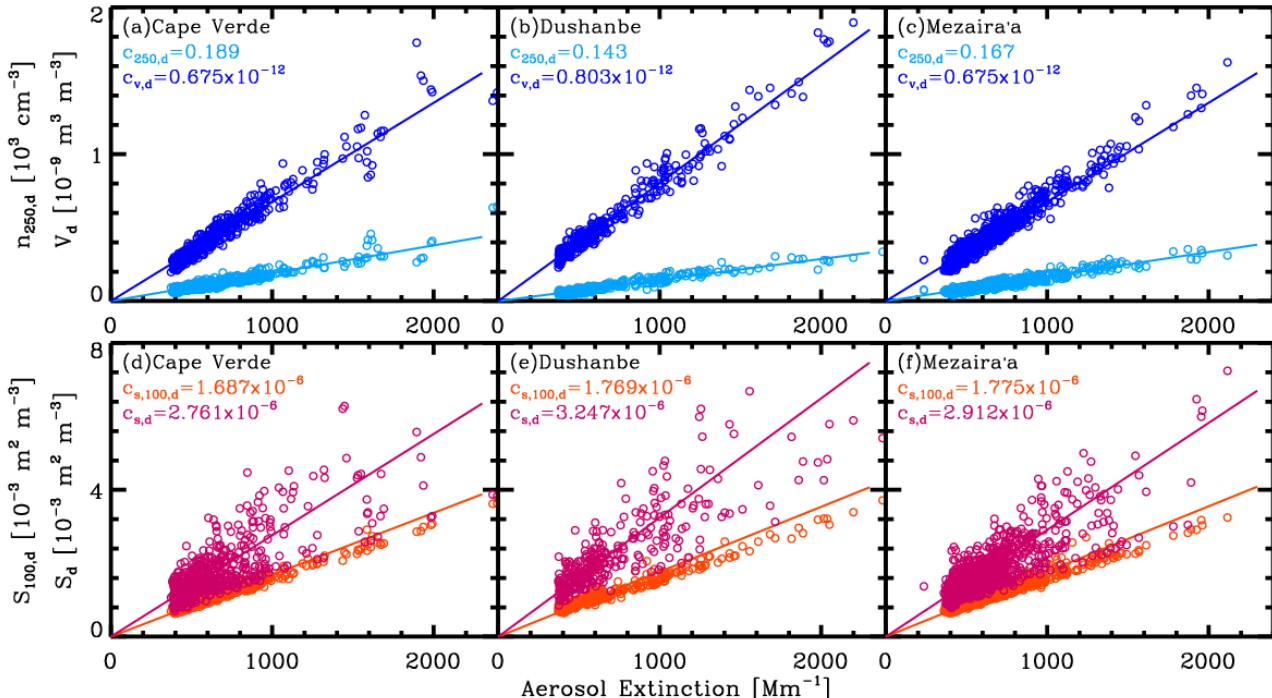

**Figure 3: Relationship between aerosol extinction coefficient at 532 nm and large particle (with radius >250 nm) number concentration $n_{250,d}$ and volume concentration $v_d$, and surface area concentration $s_d$ and $s_{100,d}$ (only considering particles with radius >100 nm) for pure dust at three typical sites, i.e., (a) and (d) for Cape Verde, (b) and (e) for Dushanbe, and (c) and (f) for Merzaira'a. The pure dust data points are selected from the AERONET V3 database (level-2.0 AOD products and level-2.0 aerosol inversions) using the threshold of particle linear depolarization ratio derived with the method given by Shin et al. (2019). The corresponding dust-related conversion factors $c_{v,d}$, $c_{250,d}$, $c_{s,d}$ and $c_{s,100,d}$ are also given, respectively.**

**Table 3. POLIPHON dust-related conversion factors $c_{v,d}$ ($10^{-12}$ Mm), $c_{250,d}$ (Mm cm$^{-3}$), $c_{s,d}$ ($10^{-12}$ Mm m$^2$ cm$^{-3}$) and $c_{s,100,d}$ ($10^{-12}$ Mm m$^2$ cm$^{-3}$) for DDM+PD and only PD, respectively. The respective standard deviations are also provided. The sites are classified into three regional clusters, including North Africa, the Middle East, and Asia.**

| | Site | $C_{v,d}$ ($10^{-12}$Mm) | | $C_{250,d}$ (Mm cm$^{-3}$) | | $C_{s,d}$ ($10^{-12}$Mm m$^2$cm$^{-3}$) | | $C_{s,100,d}$ ($10^{-12}$Mm m$^2$cm$^{-3}$) | |
| --- | --- | --- | --- | --- | --- | --- | --- | --- | --- |
| | | DDM+PD | PD | DDM+PD | PD | DDM+PD | PD | DDM+PD | PD |
| **North Africa** | CV | 0.67±0.07 | 0.68±0.06 | 0.19±0.03 | 0.19±0.03 | 2.81±0.74 | 2.76±0.69 | 1.73±0.20 | 1.69±0.13 |
| | DK | 0.67±0.09 | 0.68±0.08 | 0.18±0.03 | 0.18±0.03 | 3.01±0.84 | 2.87±0.74 | 1.83±0.30 | 1.74±0.18 |
| | IZ | 0.64±0.05 | 0.64±0.05 | 0.20±0.02 | 0.20±0.02 | 2.34±0.44 | 2.34±0.44 | 1.59±0.09 | 1.59±0.09 |
| **Middle East** | EI | 0.62±0.10 | 0.66±0.07 | 0.16±0.04 | 0.18±0.03 | 3.13±1.04 | 2.40±0.55 | 2.28±0.78 | 1.64±0.12 |
| | SV | 0.72±0.08 | 0.74±0.07 | 0.16±0.02 | 0.16±0.02 | 2.91±0.71 | 2.78±0.66 | 1.85±0.35 | 1.66±0.15 |
| | ME | 0.66±0.08 | 0.68±0.07 | 0.16±0.03 | 0.17±0.02 | 3.19±0.74 | 2.91±0.60 | 2.09±0.47 | 1.78±0.17 |
| **Asia** | DU | 0.79±0.10 | 0.80±0.07 | 0.13±0.03 | 0.14±0.02 | 3.37±0.72 | 3.25±0.71 | 2.02±0.35 | 1.77±0.19 |
| | DA | 0.68±0.16 | 0.87±0.37 | 0.18±0.04 | 0.18±0.05 | 3.46±1.40 | 2.96±1.20 | 2.19±0.70 | 1.89±0.56 |
| | LA | 0.68±0.12 | 0.74±0.10 | 0.18±0.04 | 0.16±0.02 | 3.47±0.79 | 3.35±0.78 | 1.91±0.27 | 1.69±0.15 |


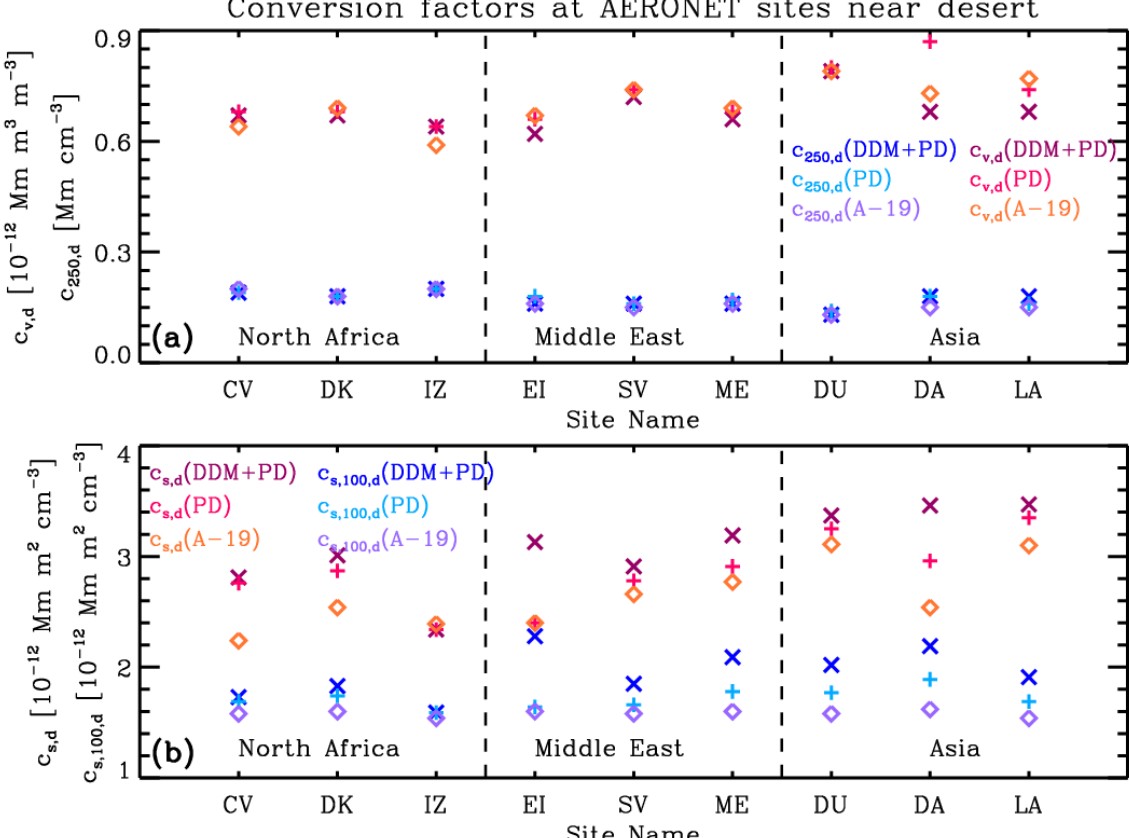

**Figure 4:** Intercomparison of dust-related conversion factors at nine sites (as shown in Table 2) near the deserts (a) $c_{v,d}$ and $c_{250,d}$, and (b) $c_{s,d}$ and $c_{s,100,d}$ calculated with the dust (both PD+DDM and PD) data selection scheme (based on particle linear depolarization ratio) in this study with those derived with the constraints of Ångström exponent for the 440–870 nm wavelength range $AE_{440-870}$ <0.3 and aerosol optical depth at 532 nm $AOD_{532}$ >0.1 given by Ansmann et al. (2019b) (denoted as 'A-19').

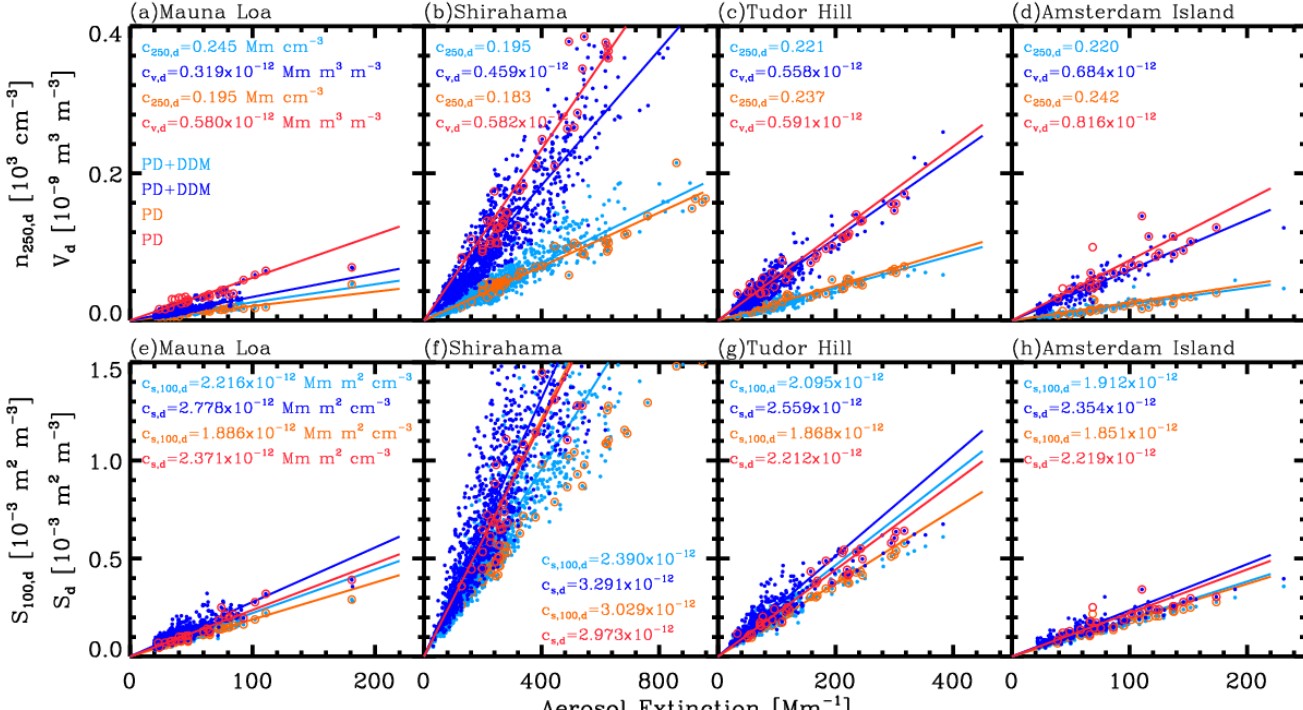

**Figure 5: Relationship between the aerosol extinction coefficient at 532 nm and large particle (radius >250 nm) number concentration $n_{250,d}$ and volume concentration $v_d$, and surface area concentration $s_d$ () and $s_{100,d}$ (radius >100 nm) for DDM+PD (in light blue and blue) and PD (in orange and red) at four typical ocean and coast sites, i.e., (a) and (d) for Mauna Loa, (b) and (f) for Shirahama, (c) and (g) for Tudor Hill, and (d) and (h) for Amsterdam Island. The PD and DDM data points are selected from the AERONET Version 3 database (level-2.0 AOD products and level-1.5 aerosol inversion products) using the dust ratio threshold derived with the method given by Shin et al. (2019). The corresponding values of dust-related conversion factors $c_{v,d}$, $c_{250,d}$, $c_{s,d}$, and $c_{s,100,d}$ are also given, respectively.**

**Table 4.** POLIPHON dust-related conversion factors $c_{v,d}$ (in $10^{-12}$ Mm), $c_{250,d}$ (in Mm cm$^{-3}$), $c_{s,d}$ (in $10^{-12}$ Mm m$^2$ cm$^{-3}$) and $c_{s,100,d}$ (in $10^{-12}$ Mm m$^2$ cm$^{-3}$) for DDM (dust-dominated mixture) +PD (pure dust) and only PD based on the AERONET data analysis. The respective standard deviations are also provided. The sites are classified into five regional clusters, including the Pacific, Pacific coast, Atlantic, Indian Ocean, and Arctic Ocean.

| | Site | $C_{v,d}$ ($10^{-12}$Mm) | | $C_{250,d}$ (Mm cm$^{-3}$) | | $C_{s,d}$ ($10^{-12}$Mm m$^2$cm$^{-3}$) | | $C_{s,100,d}$ ($10^{-12}$Mm m$^2$cm$^{-3}$) | |
|---|---|---|---|---|---|---|---|---|---|
| | | DDM+PD | PD | DDM+PD | PD | DDM+PD | PD | DDM+PD | PD |
| Pacific | TA | 0.62±0.14 | - | 0.17±0.03 | - | 2.52±0.49 | - | 1.92±0.24 | - |
| | NR | 0.68±0.16 | 0.74±0.16 | 0.20±0.04 | 0.23±0.09 | 2.12±0.40 | 1.95±0.40 | 1.75±0.25 | 1.73±0.36 |
| | MI | 0.58±0.15 | 0.67±0.16 | 0.22±0.05 | 0.19±0.03 | 2.34±0.42 | 2.04±0.27 | 1.95±0.27 | 1.73±0.19 |
| | AS | 0.63±0.15 | 0.80±0.09 | 0.20±0.04 | 0.19±0.06 | 2.33±0.48 | 2.31±0.53 | 1.88±0.27 | 1.90±0.37 |
| | GA | 0.62±0.14 | - | 0.23±0.06 | - | 2.07±0.43 | - | 1.80±0.29 | - |
| | ML | 0.33±0.10 | 0.58±0.11 | 0.25±0.05 | 0.20±0.04 | 2.86±0.64 | 2.37±0.37 | 2.30±0.40 | 1.89±0.15 |
| Pacific Coast | HU | 0.47±0.14 | - | 0.18±0.04 | - | 3.41±0.86 | - | 2.31±0.36 | - |
| | OS | 0.51±0.10 | 0.69±0.22 | 0.20±0.04 | 0.18±0.04 | 3.48±0.87 | 3.55±1.06 | 2.42±0.47 | 2.09±0.51 |
| | SH | 0.46±0.12 | 0.58±0.10 | 0.20±0.04 | 0.18±0.03 | 3.29±0.94 | 2.97±0.70 | 2.39±0.53 | 1.86±0.21 |
| | SI | 0.33±0.07 | - | 0.22±0.04 | - | 2.84±0.65 | - | 2.16±0.32 | - |
| | TR | 0.46±0.12 | 0.59±0.09 | 0.22±0.05 | 0.21±0.04 | 2.52±0.50 | 2.39±0.65 | 2.06±0.30 | 2.02±0.59 |
| Atlantic | AG | 0.53±0.13 | 0.59±0.08 | 0.23±0.05 | 0.22±0.03 | 2.36±0.42 | 2.26±0.35 | 1.90±0.28 | 1.74±0.18 |
| | TH | 0.56±0.14 | 0.59±0.14 | 0.22±0.06 | 0.24±0.04 | 2.57±0.57 | 2.21±0.36 | 2.10±0.37 | 1.87±0.20 |
| | ST | 0.50±0.13 | - | 0.22±0.06 | - | 2.89±0.92 | - | 2.25±0.55 | - |
| Indian Ocean | MG | 0.55±0.16 | - | 0.22±0.05 | - | 2.27±0.33 | - | 1.90±0.22 | - |
| | AI | 0.69±0.15 | 0.82±0.20 | 0.23±0.07 | 0.22±0.07 | 2.41±0.58 | 2.22±0.48 | 1.98±0.42 | 1.85±0.35 |
| Arctic Ocean | NA | 0.32±0.05 | - | 0.23±0.04 | - | 3.14±0.75 | - | 2.37±0.38 | - |
| | TL | 0.29±0.04 | - | 0.24±0.04 | - | 2.89±0.46 | - | 2.37±0.32 | - |
| | OP | 0.29±0.07 | - | 0.24±0.04 | - | 2.69±0.30 | - | 2.24±0.22 | - |
| | IQ | 0.30±0.06 | - | 0.28±0.05 | - | 2.94±0.48 | - | 2.47±0.31 | - |

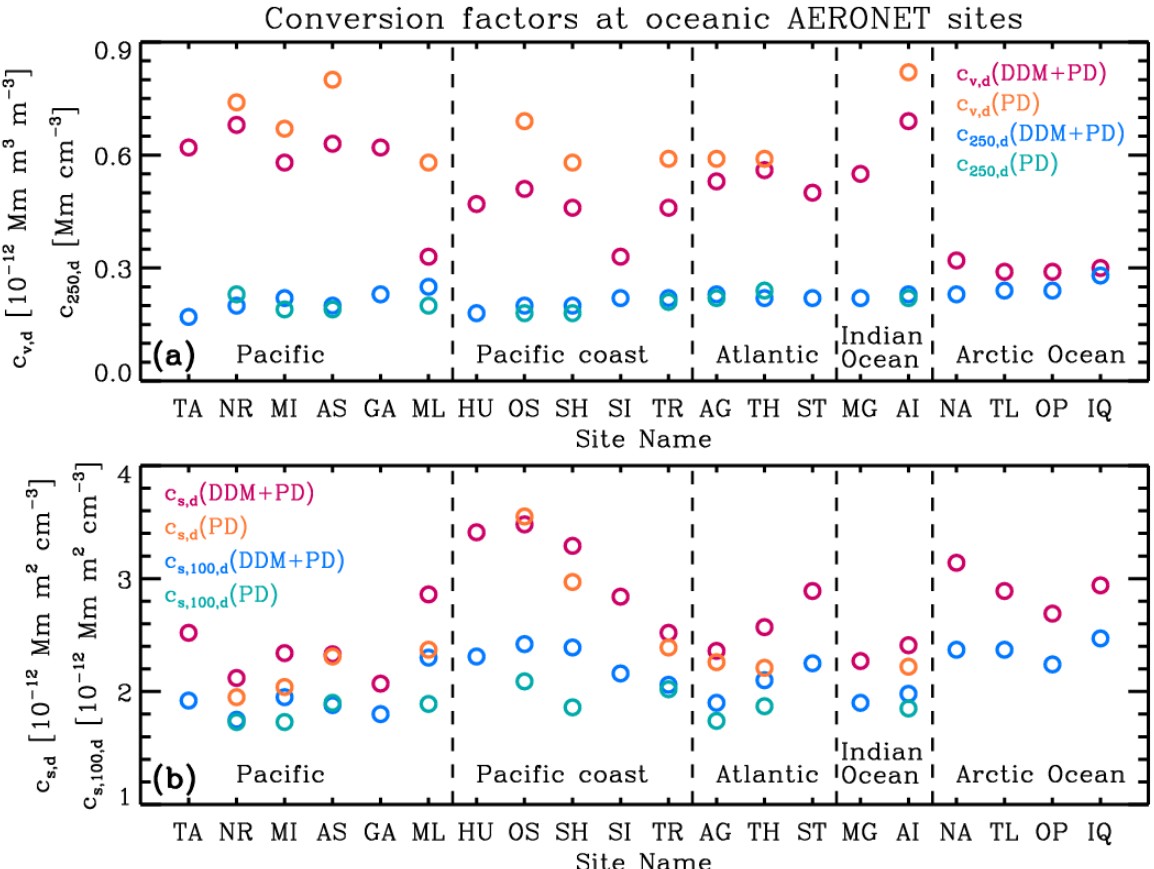

Figure 6: Dust-related conversion factors at 20 oceanic and coast sites (a) $c_{v,d}$ and $c_{250,d}$, and (b) $c_{s,d}$ and $c_{s,100,d}$ calculated by considering PD and DDM+PD datasets, respectively.

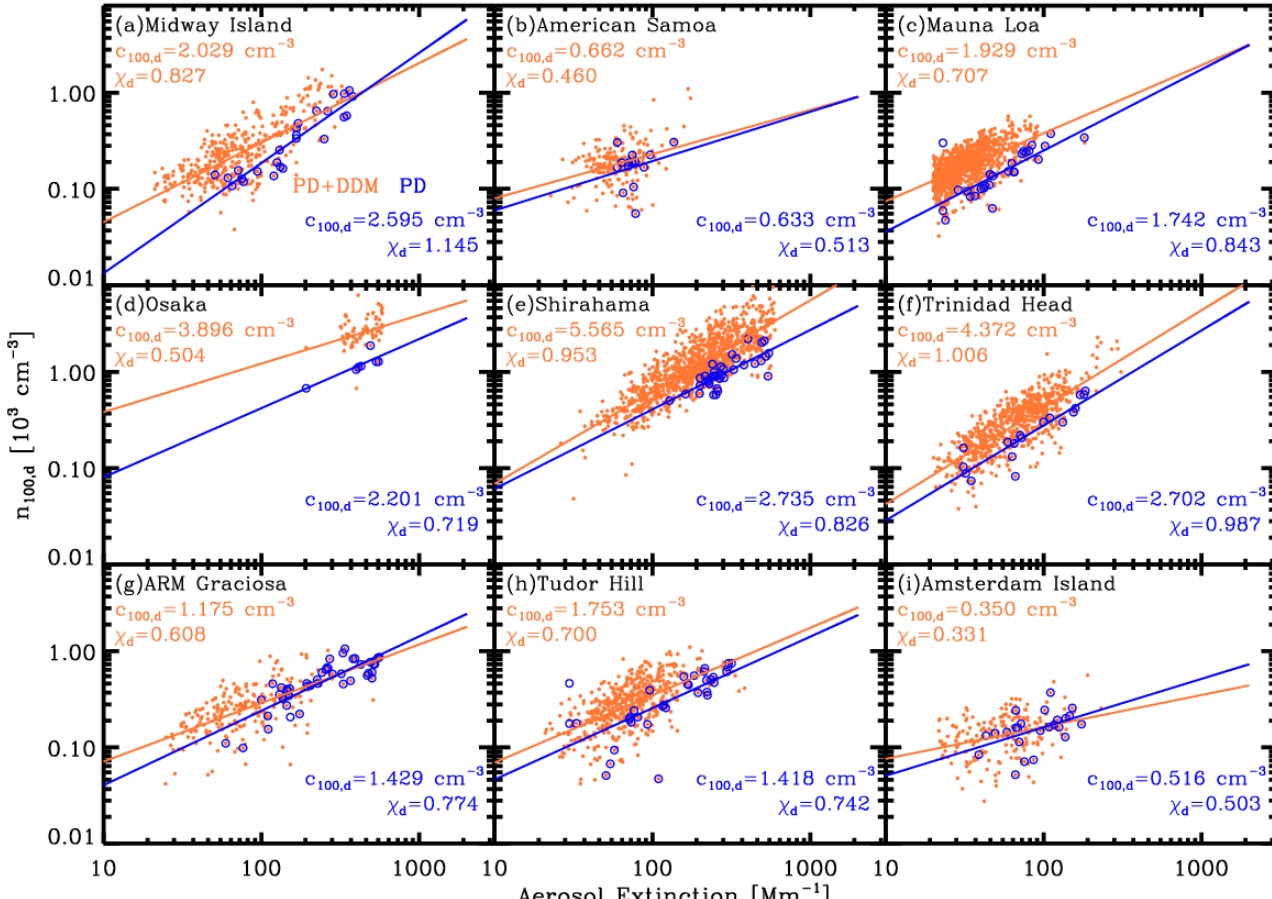

**Figure 7: Relationship between aerosol extinction coefficient at 532 nm and aerosol particle number concentration** $n_{100,d}$ **(radius >100 nm) for DDM+PD (in orange) and PD (in blue) at nine ocean and coast sites, i.e., (a) Midway Island, (b) American Samoa, (c) Mauna Loa, (d) Osaka, (e) Shirahama, (f) Trinidad Head, (g) ARM Graciosa, (h) Tudor Hill, and (i) Amsterdam Island. The PD and DDM data points are selected from the AERONET Version 3 database (level-2.0 AOD products and level-1.5 aerosol inversion products) using the dust ratio threshold derived with the method given by Shin et al. (2019). The corresponding dust-related conversion factors** $c_{100,d}$ **and** $\chi_d$ **are also provided.**

**Table 5. POLIPHON dust-related conversion factors $c_{100,d}$ (in cm$^{-3}$ for $\alpha_d$ =1 Mm$^{-1}$) and $\chi_d$ for DDM (dust-dominated mixture) +PD (pure dust) and only PD based on the AERONET data analysis. The sites are classified into 5 clusters, including the Pacific, Pacific coast, Atlantic, Indian Ocean, and Arctic Ocean.**

| Level 2.0 | Site | $C_{100,d}$ (cm$^{-3}$ for $\alpha_d$=1 Mm$^{-1}$) | | $\chi_d$ | |
|---|---|---|---|---|---|
| | | DDM+PD | PD | DDM+PD | PD |
| Pacific | Tahiti (TA) | 2.50 | - | 0.86 | - |
| | Nauru (NR) | - | - | - | - |
| | Midway_Island (MI) | 2.03 | 2.60 | 0.83 | 1.15 |
| | American_Samoa (AS) | - | 0.63 | - | 0.51 |
| | Guam (GA) | - | - | - | - |
| | Mauna_Loa (ML) | 1.93 | 1.74 | 0.71 | 0.84 |
| Pacific coast | Hokkaido_University (HU) | 5.62 | - | 0.97 | - |
| | Osaka (OS) | 3.90 | 2.20 | 0.50 | 0.72 |
| | Shirahama (SH) | 5.57 | 2.74 | 0.95 | 0.83 |
| | Saturn_Island (SI) | 3.95 | - | 0.88 | - |
| | Trinidad_Head (TR) | 4.37 | 2.70 | 1.01 | 0.99 |
| Atlantic | ARM_Graciosa (AG) | 1.18 | 1.60 | 0.61 | 0.89 |
| | Tudor_Hill (TH) | 1.75 | 1.42 | 0.70 | 0.74 |
| | St_Helena (ST) | 1.99 | - | 0.69 | - |
| Indian Ocean | Maldives_Gan (MG) | 2.65 | - | 0.88 | - |
| | Amsterdam_Island (AI) | - | 0.52 | - | 0.50 |
| Arctic Ocean | Narsarsuaq (NA) | 3.69 | - | 0.83 | - |
| | Thule (TL) | 3.93 | - | 0.82 | - |
| | OPAL (OP) | 2.04 | - | 0.60 | - |
| | Iqaluit (IQ) | 3.91 | - | 0.82 | - |

810

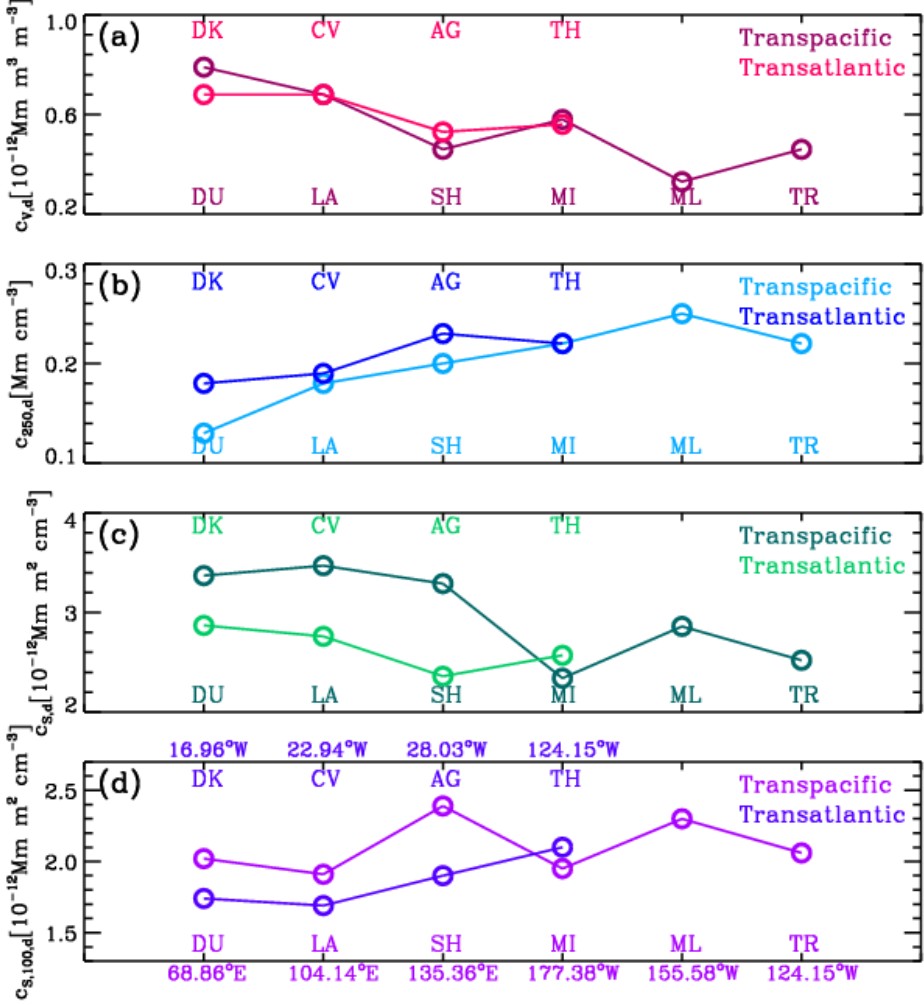

**Figure 8: Variations in conversion factors (a) $c_{v,d}$, (b) $c_{250,d}$, (c), $c_{s,d}$ and (d) $c_{s,100,d}$ along transoceanic dust transport paths, including a transpacific path from Asian dust sources to the west coast of North America and a transatlantic path from the Saharan Desert (North Africa) to the east coast of North America. The DDM and PD datasets are considered together for calculating the conversion factors. The longitudes of the sites are also shown.**