# Peer review of "POLIPHON conversion factors for retrieving dust-related cloud condensation nuclei and ice-nucleating particle concentration profiles at oceanic sites"

_Atmospheric Measurement Techniques, 2023_

## Referee Comment (RC3)

The paper "POLIPHON conversion factors for retrieving dust-related cloud condensation nuclei and ice-nucleating particle concentration profiles at oceanic sites" presents and discusses the dust-related conversion factors as extracted over remote oceanic/coast sites using the AERONET database around the globe. These different conversion parameters are of critical importance for the POLIPHON methodology in order to compute dust-related CCNC and INPC globally. The study falls within the scope of AMT. The authors have done a thorough job, the manuscript is well-written / structured, the presentation clear, the language fluent and the quality of the figures high. Furthermore, the authors give credit to related work and the results support the conclusions. However, in order to help improving the manuscript, I would kindly suggest the authors to take into account the following minor comments.

Comments:

1. Central component of the analysis provided is the AERONET-based depolarization ratio, which is established according to the model of randomly oriented spheroids. Thus, I would suggest to the authors to dive into significantly more discussion and details on the major building component of their approach, including methodology, approach, assumptions, and accuracy.
2. Since a significant number of PollyXT lidars operate at same time AERONET stations, my suggestion would be the extended intercomparison and evaluation of the AERONET-based depolarization ratio against the Polly lidar depolarization ratio, under events of dust, polluted dust, and non-dust, in order to strengthen the argument of the suitability of the AERONET-based depolarization ratio to extract CCNC and INPC conversion factors. This comparison over land should be a first stepping stone before attempting over ocean, where lidar systems are less frequently operated, and eventually before the claim of supporting 3D CCNC and INPC dust-related studies globally.
3. The authors should go into more details on the dependencies between the AERONET-based depolarization ratio to extract CCNC and INPC conversion factors around the globe and the discrepancies in dust microphysical properties of dust around the globe, for the main objective is to apply the conversion factors eventually in lidar observations through POLIPHON, possible at regions and conditions of dust transport significantly different than the observed at the specific stations of the present study. Moreover, the authors should discuss, possible through study cases, the change of the extracted and proposed CCNC and INPC conversion factors as a function of aeolian transport and distance, for aging and mixing with non-dust aerosol subtypes, even under the hypothesis of external mixing, alters the columnar observations of AERONET, thus affects the total conversion factors.
4. The aforementioned approach should be as robust a possible, for once the conversion factors are extracted and established for the dust CCNC and INPC over a region, the product output should consist a fingerprint of the dust related sources affecting the region as well, interconnecting the dust plumes over the oceanic sites with the dust sources. Towards this, I would suggest the authors to perform a cluster analysis of the dust sources affecting each oceanic site (i.e., backtrajectories).

5. Please discuss the effect on the extracted AERONET-based depolarization ratios and accordingly on the CCNC and INPC conversion factors of different dust regions – with different dust properties (i.e. LR), affecting the same marine site.

6. Table 2 provides the available number of data points for total, dust-dominated mixture, and pure dust, in AERONET inversion products. In specific cases the dataset is characterized by a very low number of cases. The authors should discuss on the fail-safes considered in order to guarantee the robustness of the conversion factors extracted, even in the low number of cases AERONET stations. Moreover, please provide at the table for each of the site (Table 2), with the basis AERONET products, for the Total Obs., DDM Obs., and PD Obs. (i.e., AOD+AOD_SD, AE+AE_SD, …). How does the low number of cases affect the uncertainties and confidence of the conversion factors? Please discuss providing additional input where necessary.

7. In table 1 the authors provide the uncertainties of the approach. The uncertainties have been established on the basis of long-term ground-based observations (i.e., EARLINET, PollyNET). Since the objective of the study, as mentioned in the very beginning of the manuscript, is to "to characterize the 3-D distribution of dust-related Cloud Condensation Nuclei Concentration (CCNC) and Ice Nucleating Particle Concentration (INPC) globally", which can be achieved only based on satellite-lidar systems (i.e., CALIOP, CATS, ATLID), where the uncertainties of backscatter and particulate depolarization ratio are of the same order of magnitude as the backscatter and particulate depolarization ratio. In this case, as the higher uncertainties are used as input in the error propagation, the final uncertainties will be significantly higher in satellite-based lidar products than when ground based products are extracted. Please discuss.

---

## Author Comment (AC2)

**Response to RC1**

**General Comments:**

The manuscript presents a new set of POLIPHON conversion factors for dust aerosols at oceanic or coastal sites. The authors use a depolarization ratio-based metric to identify dust cases in AERONET measurements and further classify them into clusters of pure dust (PD) and dust-dominated mixture (DDM). They estimate the CCN- and INP-related conversion factors for these two clusters and also compare them with the already existing ones. In addition, they discuss the variations of these conversion factors along the transoceanic pathways. The manuscript presents considerable development in the field of lidar-based CCN and INP retrieval and has good potential for publication in AMT only after implementing and addressing the following comments.

**Response:** We appreciate the reviewer's thoughtful review and constructive comments. All the comments have been addressed in the revised manuscript, and the responses to each comment are given below.
* * *
**Specific comments:**

**Comments:** Since you mention the significance of how estimating the 3D distribution of CCN and INP can improve the climate models twice, in lines 52–54 and 71–72, could you please elaborate in the text with one or two sentences on how it may do that?

**Response:** According to the reviewer's suggestion, we have added related sentences respectively for the mentioned two places in the revised manuscript as follows:

**'Phillips et al. (2013) stated that the reliable quantification of the linkage between aerosol conditions and ice crystal numbers should be the first step in quantifying cold-cloud indirect effects. Thus, INP concentrations (INPC) are estimated in many studies to predict the initial in-cloud ice crystal number concentration (ICNC) via primary heterogeneous nucleation (Ansmann et al., 2019a; He et al., 2022a; Kanji et al., 2017). Moreover, discrepancies between INPC and ICNC are found to establish the role of secondary ice nucleation (DeMott et al., 2011). Therefore, the constraint of ambient INPC can…'** (Please see Line 50-55)

**'Therefore, the number concentration of cloud drops and ice crystals can be well quantified (Ramanathan et al., 2001), thus improving the current consideration of ACIs….'** (Please see Line 76-77)

**Comments:** The microphysical properties of dust may change with ageing and mixing with other aerosol types (Kim and Park, 2012; Goel et al., 2020). This implies that the conversion factors estimated close to the main desert sources may not be the same as those estimated far away from the source region. This may be stated in the motivation.

**Response:** We have rephrased the related sentences as below and the suggested papers have been cited now. **'The dust-related conversion factors can be very different for the downstream areas far from the dust sources due to the possible aging and mixing of dust with other aerosol types during long-range transport (Kim and Park, 2012; Goel et al., 2020). However, for these downstream areas, dust-related conversion factors are still lacking owing to insufficient data points fulfilling the criteria.'** (Please see Line 88-91)

**Comments:** Some validation studies that use POLIPHON conversion factors to estimate aerosol number concentrations and CCN concentrations from spaceborne lidar measurements highlight the need for improvement, especially for dust and marine aerosol mixtures (Choudhury et al., 2022; Choudhury and Tesche, 2022). Such studies can be included in the motivation for this study.

**Response:** Thank you for the reviewer's suggestion. We have added a sentence to mention this motivation as below **'The POLIPHON method has been proven to be useful for examining the profiles of CCNC, INPC, and aerosol number concentration retrieved from spaceborne lidar measurements with other algorithms (Choudhury and Tesche, 2022a, 2022b; Choudhury et al., 2022c).'** (Please see Line 62-64)

**Comments:** Line 136. Ansmann et al. (2019) use aerosol optical thickness (AOT) at 532 nm, not 500 nm. They estimate AOD at 532 nm using Ångström exponent and AOT at 500 nm. Since you compare your results with Ansmann et al. (2019), it is necessary to stay consistent. Also, 532 nm is the usual lidar wavelength. As you aim for application to spaceborne lidar, you should use this wavelength in your analysis. In general, the wavelengths mentioned in the paper for AOT and extinction coefficient up to this point are inconsistent. For instance, it is 532 nm in line 60 and 550 nm in line 139. Did you convert the AOT from 500 nm to 532 nm to estimate the conversion factors? If not, then how do you justify your calculations, as we cannot ignore the wavelength dependency of optical thickness?

**Response:** In the revised manuscript, to be consistent with the calculation in Ansmann et al. (2019), we have updated the dust-related conversion factors by converting AOD at 500nm into AOD at 532nm using Ångström exponent. It should be noted that only very small changes are found for the obtained conversion factor values. Also, '550 nm' in line 139 has been modified to '**532 nm**'.

**Comments:** Line 218. It seems like you did not convert the AOT to 532 nm when calculating the conversion factors. Please address the previous comment.

**Response:** We have converted AOD at 500 nm into AOD at 532 nm by considering the

Ångström exponent. Therefore, all the dust-related conversion factors are updated as seen in Figure 3, Table 3, Figure 4, Figure 5, Table 4, Figure 6, Figure 7, Table 5, Figure 8, and Figure A1. It should be noted that only very small changes are found for the obtained conversion factor values.

**Comments:** I found the transition from Section 2 to Section 3 to be very abrupt. I suggest adding a paragraph under Section 3 before going to Section 3.1, giving a brief introduction to the section and the analysis you present here so that readers have a basic idea of what to expect. This should have been included in the "Data and Methodology" section. However, looking at the organisation of the manuscript, I suggest adding it at the beginning of Section 3.

**Response:** Thank you for pointing out this. Taking the reviewer's suggestion into consideration, we have added a new paragraph at the beginning of section 3 (before section 3.1) in the revised manuscript as follows '**In this section, we mainly focus on the calculation of the dust-related conversion factors in the POLOPHON method with the new dust identification scheme, which is based on the particle linear depolarization ratio in the AERONET data product. To verify the performance of the proposed dust identification scheme, the dust-related conversion factors near deserts are first calculated at nine AERONET sites and compared with those obtained by Ansmann et al. (2019b). Then, the dust-related conversion factors $c_{v,d}$, $c_{250,d}$, $c_{s,d}$, $c_{s,100,d}$, $c_{100,d}$, and $\chi_d$ at 20 oceanic/coastal AERONET sites are derived with the proposed method. Finally, the variations in the dust-related conversion factors along the two transoceanic (i.e., transatlantic and transpacific) pathways are analyzed.**' (Please see Line 232-238)

**Comments:** Lines 226-228. How can the number of data samples be linked to the level of agreement or difference between pure dust (PD) and PD + dust-dominated mixtures (DDM) clusters? Having fewer samples could only imply that the result is not significant or lacks confidence.

**Response:** For accuracy, we have removed this sentence.

**Comments:** Line 231. Can you speculate on the local aerosol sources based on the sites or locations where you found the differences between PD and DDM clusters?

**Response:** We preliminarily checked the data from other terrestrial AERONET sites across the world with sufficient dust cases available. These strange results are only found for some sites in the following three regions: (1) Middle East (e.g., KAUST_Campus, Bahrain, Dhadnah, Solar_Village, Weizmann_Institute, and SEDE_BOKER); (2) Africa (Blida, Cairo_EMA_2, Dakar, IER_Cinzana, Ilorin, and Tunis_Carthage); (3) polluted Europe sites (ATHENS-NOA, Coruna, CUT-TEPAK,

El_Arenosillo, FORTH_CRETE, Granada, IMS_METU_ERDEMLI, Lampedusa, Palma_de_Mallorca, Pairs, and Rome_Tor_Vergata). The strange pattern for the DDM cluster can even extend at the extinction coefficient range of 100-800 Mm$^{-1}$, such as at Cairo_EMA_2 and Solar_Village. When we remove the DDM data from DDM+PD data, the Pearson correlation coefficient for $C_{s,d}$ and $C_{s,100,d}$ will significantly increase, for example, from 0.7-0.8 to >0.95 (this is the usual situation for most sites as we checked).

Considering that this strange pattern appears at the sites from different regions, it would be difficult to give credible speculation on the local aerosol sources. This strange pattern does not appear at oceanic sites that are focused on in this study. Here we mention this strange pattern so as to remind that more care should be taken when employing DDM data to retrieve conversion factors at terrestrial sites in the Middle East, Africa, and polluted European cities. Therefore, to remind this, we have added a sentence in the revised manuscript as below '**Thus, it should be noted that more care should be taken when employing DDM data to retrieve dust-related conversion factors at terrestrial sites in the Middle East, Africa, and polluted European cities in future work.**' (Please see Line 260-261)

**Comments:** The comparisons presented in Section 3.1 are all qualitative. I would always prefer and recommend a quantitative analysis. Please quantify the differences in terms of percentages or absolute values.

**Response:** Considering the reviewer's suggestion, we have updated the text of the third paragraph of section 3.1 by using the percentages to describe the differences in conversion factors between A-19 and this study. (Please see Line 265-270)

**Comments:** Line 260. PD cases are already included in PD+DDM clusters, right? Please change "PD and PD+DDM" to "PD and DDM". Also, how do you define an adequate sample size? Please include it in the text.

**Response:** Yes, PD cases are included in PD+DDM clusters. In Ansmann et al. (2019b), the lowermost number of available data points for calculating the conversion factors is 17 at the Tuscon AERONET site. In this study, we provide INP-relevant ($c_{v,d}$, $c_{250,d}$, $c_{s,d}$, and $c_{s,100,d}$) conversion factors (see tables 3 and 4) with the number of available data points no less than 12 to ensure as many sites as possible can provide dust-related conversion factors with somewhat acceptable reliability. With data point number below 12, the data points may be diverging caused by occasional dust cases, causing very small linear Pearson correlation coefficients R. Here we provide R values for each dust-related INP-relevant conversion factor at each AERONET employed in this study as seen in the following table. It can be seen clearly that most of the conversion factors have corresponding R values exceeding 0.70, except for PD-derived $c_{250,d}$ values at NR and AS (as marked in red in the table), which can guarantee the robustness of the

retrieved conversion factors.

| | Site | R for $C_{v,d}$ | | R for $C_{250,d}$ | | R for $C_{s,d}$ | | R for $C_{s,100,d}$ | |
|---|---|---|---|---|---|---|---|---|---|
| | | DDM+PD | PD | DDM+PD | PD | DDM+PD | PD | DDM+PD | PD |
| **North Africa** | CV | 0.97 | 0.97 | 0.94 | 0.94 | 0.76 | 0.78 | 0.98 | 0.99 |
| | DK | 0.96 | 0.96 | 0.94 | 0.94 | 0.71 | 0.74 | 0.94 | 0.97 |
| | IZ | 0.98 | 0.98 | 0.92 | 0.92 | 0.77 | 0.77 | 0.99 | 0.99 |
| **Middle East** | EI | 0.95 | 0.97 | 0.91 | 0.92 | 0.56 | 0.82 | 0.56 | 0.99 |
| | SV | 0.97 | 0.98 | 0.96 | 0.96 | 0.76 | 0.78 | 0.91 | 0.98 |
| | ME | 0.96 | 0.98 | 0.93 | 0.96 | 0.76 | 0.82 | 0.79 | 0.97 |
| **Asia** | DU | 0.98 | 0.98 | 0.95 | 0.96 | 0.84 | 0.83 | 0.94 | 0.98 |
| | DA | 0.90 | 0.77 | 0.76 | 0.83 | 0.41 | 0.93 | 0.67 | 0.70 |
| | LA | 0.96 | 0.97 | 0.86 | 0.93 | 0.76 | 0.73 | 0.95 | 0.99 |
| **Pacific** | TA | 0.88 | - | 0.89 | - | 0.87 | - | 0.94 | - |
| | NR | 0.93 | 0.80 | 0.90 | 0.32 | 0.91 | 0.79 | 0.96 | 0.76 |
| | MI | 0.92 | 0.91 | 0.96 | 0.97 | 0.95 | 0.97 | 0.97 | 0.98 |
| | AS | 0.89 | 0.91 | 0.84 | 0.50 | 0.86 | 0.75 | 0.92 | 0.81 |
| | GA | 0.80 | - | 0.86 | - | 0.89 | - | 0.93 | - |
| | ML | 0.88 | 0.92 | 0.92 | 0.95 | 0.90 | 0.95 | 0.95 | 0.99 |
| **Pacific Coast** | HU | 0.84 | 1.00 | 0.84 | - | 0.87 | - | 0.93 | - |
| | OS | 0.86 | 0.90 | 0.75 | 0.98 | 0.66 | 0.79 | 0.76 | 0.87 |
| | SH | 0.93 | 0.99 | 0.94 | 0.97 | 0.89 | 0.88 | 0.90 | 0.98 |
| | SI | 0.92 | - | 0.94 | - | 0.89 | - | 0.96 | - |
| | TR | 0.88 | 0.98 | 0.93 | 0.98 | 0.93 | 0.97 | 0.95 | 0.98 |
| **Atlantic** | AG | 0.98 | 0.99 | 0.98 | 0.98 | 0.95 | 0.97 | 0.99 | 0.98 |
| | TH | 0.93 | 0.98 | 0.92 | 0.97 | 0.92 | 0.97 | 0.96 | 0.99 |
| | ST | 0.91 | - | 0.91 | - | 0.95 | - | 0.98 | - |
| **Indian Ocean** | MG | 0.82 | - | 0.83 | - | 0.92 | - | 0.95 | - |
| | AI | 0.93 | 0.82 | 0.89 | 0.82 | 0.93 | 0.85 | 0.98 | 0.92 |
| **Arctic Ocean** | NA | 0.92 | - | 0.93 | - | 0.76 | - | 0.88 | - |
| | TL | 0.95 | - | 0.91 | - | 0.91 | - | 0.95 | - |
| | OP | 0.76 | - | 0.94 | - | 0.87 | - | 0.91 | - |
| | IQ | 0.74 | | 0.72 | - | 0.84 | - | 0.88 | - |

For clarity, we have rephrased this sentence in the revised manuscript as '**The results for only PD cluster and combined PD and DDM clusters are listed. We consider the conversion factors with ≥12 available PD data points valid (provided in Table 4). Moreover, to guarantee robustness, only the retrieved conversion factors with the linear Pearson correlation coefficient $R$ exceeding 0.70 are considered valid, except for PD-derived $c_{250,d}$ values at NR (R=0.32) and AS (R=0.50), which should especially be handled with care in scientific applications.**' (Please see Line 288-292)

**Comments:** Line 306. You haven't mentioned the cluster you used in Figure 8. Looking at the values, I guess it is for PD cases. Here, I am assuming that the variations in conversion factors along the transoceanic pathways are for PD cases. From Table 2, I can see that the number of samples for stations at MI, ML, and TR are 26, 27, and 18, which are significantly less than other stations. Due to fewer samples, the variations that you report for these stations may not be realistic, as the long-range dust transports may vary seasonally and annually. One thing to look at is the distribution of the small number of sample points across different seasons and years. Are they limited to one season, or one year? In any case, you must mention these limitations in the paper. I would recommend using the DDM cluster, which has more than enough sample space to study the changes in the conversion factors along different transport pathways.

**Response:** Thank you for pointing out this issue. According to the reviewer's suggestion, we have updated figure 8 by using the conversion factors calculated from the PD+DDM cluster so as to make more data points available. Now, the results may be more reliable and the related statements have been updated accordingly. (Please see Line 357, 364-367)

As for the seasonal and annual variations of the characteristics of transoceanic dust, however, the sample numbers of dust cases are still too small to support us in conducting this analysis. To our point of view, according to the dust activity in the dust sources, the transpacific dust transport from East Asia to the west coast of North America mainly occurs in spring and the transatlantic dust transport from North Africa to the east coast of North America mainly occurs in summer. Thus, seasonal variation in the dust microphysical properties (determining the dust-related conversion factors) may be not significant. Moreover, the annual variation is somewhat out of the scope of this manuscript; we would like to obtain the general conversion factors to reflect the multi-year (please see the period of data given in table 2) average feature.

**Comments:** Why did you exclude the transoceanic variations of $c_{100,d}$? I recommend adding a new panel to Figure 8 to show the variations of $c_{100,d}$, and $x_d$ and discussing them in Section 3.4 of the manuscript.

**Response:** Thank you for pointing out this issue. After careful checking, it fails to give

the results of $c_{100,d}$ at the four sites (i.e., DK, CV, DU, and LA) before transoceanic transport. The regression coefficients $\chi_d$ are found to be far less than 0.5 for these sites. Thus, it should be mentioned that using our method to retrieve the CCN-relevant conversion factors seems not robust on the continent. Therefore, we would like not to add the CCN-relevant conversion factors in figure 8. Here we have added some sentences at the end of section 3.3 to discuss the possible problem when retrieving the CCN-relevant conversion factors on the continent as follows '**Moreover, it should be mentioned that using the newly-proposed dust dataset selection scheme to retrieve the CCN-relevant conversion factors seems not robust on the continent. Thus, more care should be taken when retrieving $c_{100,d}$ and $\chi_d$ for those polluted city regions in future work.**' (Please see Line 343-345)

**Comments:** As highlighted in the introduction of the manuscript, the ultimate goal is to apply the conversion factors to spaceborne lidar measurements to estimate global 3D CCN or INP data. How do you suggest applying these new conversion factors to spaceborne lidar? Should it be applied based on the geographical location? Or, as suggested by Ansmann et al. (2019), global average conversion factors should be used?

**Response:** As mentioned in this manuscript, there will be a following-up work that focuses on the dust-related conversion factors at those polluted city sites and then the global conversion factor dataset can be expected. AERONET sites are serried in some regions (especially the regions with large populations) and can be very sparse in those outlying regions. Thus, the retrieved conversion factors absolutely will be distributed unevenly. Besides the possible use of the global average conversion factor value suggested by Ansmann et al. (2019), we will try to ensure that there are at least regional-representative conversion factors available for most of the geographical locations around the world. A dust-related conversion factor at an isolated site can be applied to a large area around it. Also, geographical interpolation is another possible way to obtain the final global grid dataset of dust-related conversion factors. Nevertheless, the final processing for retrieving the dust-related global conversion factors will be determined only after finishing the following-up work (i.e., conversion factors at polluted city sites), which would be better discussed in detail in our next paper. Here, we have added a sentence to preliminarily explain this issue in the last paragraph of the revised manuscript '**Once those conversion factors at polluted city sites are retrieved, a global dust-related conversion factor grid dataset will be obtained possibly by geographical interpolation.**' (Please see Line 397-398)

**Comments:** Figure 1 is missing some sites that are included in Table 2. Please modify.

**Response:** We have updated figure 1 in the revised manuscript to include nine near-desert sites for comparison with Ansmann et al. (2019).

**Comments:** Table 1 has an expression for $n_{CCN}$ but lacks the expression for $n_{100,d}$. Please add.

**Response:** An expression has been added in table 1 as below
$$n_{100,d}(z) = c_{100,d} \times \alpha_d(z)^{\chi_d}$$
* * *
**Technical corrections:**

**Comments:** Line 41. Please cite the latest IPCC report. And since you quoted the IPCC report, I believe you mean effective radiative forcing and not radiation budget.

**Response:** 'IPCC 2013' has been replaced by '**IPCC 2021**'. 'radiation budget' has been replaced by '**effective radiative forcing**'.

**Comments:** Line 60. A two-step dust separation technique for obtaining fine and coarse mode contributions separately, given by Mamouri and Ansmann (2014), has also been used in multiple studies and can be included here.

**Response:** We have added the following statement '**…as well as from fine and coarse mode components (Mamouri and Ansmann, 2014)**'

**Comments:** Line 74. Replace "regional-dependent and relevant to" with "regionally variable and dependent on".

**Response:** 'regional-dependent and relevant to' has been modified to '**regionally variable and dependent on**'.

**Comments:** Lines 81-82. The term "dust transport pathways" appears for the first time here without any prior explanation. I suggest explaining it briefly here.

**Response:** We have added a sentence to explain the long-range transport of dust as below '**Dust particles are frequently elevated from the surface of desert regions by wind or thermal convection and can sometimes undergo advective transport over a long range.**' (Please see Line 87-88)

**Comments:** Line 89. Do you mean "depolarization ratio"?

**Response:** Thank you for pointing out the mistake. 'polarization ratio' has been modified to '**depolarization ratio**'.

**Comments:** Lines 87-88. I suggest replacing "we plan to adopt another scheme to select

data points that are representative of dust presence from AERONET databases" with "a different scheme to identify the presence of dust in AERONET measurements".

**Response:** According to the reviewer's suggestion, this sentence has been revised as '**…, we use a different scheme to identify the presence of dust in AERONET measurements.**'

**Comments:** Lines 101-102. Replace "a previous study" with "the results from Ansmann et al. (2019)".

**Response:** 'a previous study' has been modified to '**the results from Ansmann et al. (2019b)**'.

**Comments:** Line 138. Replace "AERONRT" with "AERONET".

**Response:** 'AERONRT' has been replaced by '**AERONET**'.

**Comments:** Rephrase lines 201-202.

**Response:** This sentence has been rewritten as '**The column-integrated dust ratio ($R_{d,1020}$), representing the contribution proportion of dust backscatter to the total particle backscatter in the atmospheric column, is defined as follows: …**' (Please see Line 219-220)

**Comments:** Line 208. The abbreviation FMF appears for the first time.

**Response:** 'fine-mode fraction' has been added before the first use of '**FMF**'.

**Comments:** Line 209. A flow chart of what?

**Response:** We have added '**for dust-occurring data point selection and dust-related conversion factors retrieval**'. (Please see Line 227-228)

**Comments:** Line 241. AERONET.

**Response:** 'AERONRT' has been replaced by '**AERONET**'.

**Comments:** Line 249. Remove "scatters regarding the".

**Response:** '**scatters regarding the**' has been removed.

**Comments:** Line 250. There's no need to mention the colours here. This should be included in the figure caption. Replace "situations" with "cases".

**Response:** The colors have been moved to the caption of Figure 5 in the revised manuscript. 'situations' has been replaced by '**cases**'.

**Comments:** Line 253. Replace "participate" with "are considered".

**Response:** 'participate' has been modified to '**are considered**'.

**Comments:** There were many other obvious language-related errors that I have not included here. Some of them can be corrected during the manuscript's copyediting if it reaches that stage. However, I highly recommend the authors consult a professional language editor to improve the readability of the manuscript.

**Response:** Thank you for the review's suggestion. For readability, the language of the revised manuscript has been polished by a professional English language editing service provided by 'American Journal Experts'.
* * *
**References:**

Ansmann, A., Mamouri, R.-E., Hofer, J., Baars, H., Althausen, D., and Abdullaev, S. F.: Dust mass, cloud condensation nuclei, and ice-nucleating particle profiling with polarization lidar: updated POLIPHON conversion factors from global AERONET analysis, Atmos. Meas. Tech., 12, 4849–4865, https://doi.org/10.5194/amt-12-4849-2019, 2019.

Choudhury, G., Ansmann, A., and Tesche, M.: Evaluation of aerosol number concentrations from CALIPSO with ATom airborne in situ measurements, Atmos. Chem. Phys., 22, 7143–7161, https://doi.org/10.5194/acp-22-7143-2022, 2022.

Choudhury, G., Tesche, M.: Assessment of CALIOP-Derived CCN Concentrations by In Situ Surface Measurements, Remote Sensing, 14(14), 3342. https://doi.org/10.3390/rs14143342, 2022.

Goel, V., Mishra, S. K., Pal, P., Ahlawat, A., Vijayan, N., Jain, S., and Sharma, C.: Influence of chemical aging on physico-chemical properties of mineral dust particles: a case study of 2016 dust storms over Delhi, Environ. Pollut., 267, 115338, https://doi.org/10.1016/j.envpol.2020.115338, 2020.

Kim, J. S. and Park, K.: Atmospheric aging of Asian dust particles during long range transport, Aerosol Sci. Technol., 46, 913–924, https://doi.org/10.1080/02786826.2012.680984, 2012.

Mamouri, R. E. and Ansmann, A.: Fine and coarse dust separation with polarization lidar, Atmos. Meas. Tech., 7, 3717–3735, https://doi.org/10.5194/amt-7-3717-

2014, 2014.

DeMott, P. J., Möhler, O., Stetzer, O., Vali, G., Levin, Z., Petters, M. D., Murakami, M., Leisner, T., Bundke, U., Klein, H., Kanji, Z. A., Cotton, R., Jones, H., Benz, S., Birkmann, M., Rzesanke, D., Saathoff, H., Nicolet, M., Saito, A., Nillius, B., Bingemer, H., Abbatt, J., Ardon, K., Ganor, E., Georgakopoulos, D. G., and Saunders, C.: Resurgence in Ice Nuclei Measurement Research, B. Am. Meteorol. Soc., 92, 1623–1635, https://doi.org/10.1175/2011BAMS3119.1, 2011.

Kanji, Z. A., Ladino, L. A., Wex, H., Boose, Y., Burkert-Kohn, M., Cziczo, D. J., and Krämer, M.: Overview of ice nucleating particles, Meteor. Mon., 58, 1.1–1.33, https://doi.org/10.1175/AMSMONOGRAPHS-D-16-0006.1, 2017.

Phillips, V., DeMott, P., Andronache, C., Pratt, K., Prather, K., Subramanian, R., and Twohy, C.: Improvements to an empirical parameterization of heterogeneous ice nucleation and its comparison with observations, J. Atmos. Sci., 70, 378-409, https://doi.org/10.1175/JAS-D-12-080.1, 2013.

Ramanathan, V., Crutzen, P. J., Kiehl, J. T., and Rosenfeld, D.: Aerosols, climate, and the hydrological cycle, Science, 294, 2119–2124, https://doi.org/10.1126/science.1064034, 2001.

---

## Author Comment (AC3)

**Response to RC2**

**General Comments:**

Possibility to estimate the particle concentration (and so CCN and INP) from extinction/backscattering lidar measurements by using corresponding conversion factors is quite attractive. As suggested in publications of Ansmann with co-authors, the conversion factors can be estimated from AERONET measurements and this study improves the knowledge of these factors for dust, applying POLIPHON technique for different дщсфешщты. Paper is well written, presents new results and is suitable for publishing in AMT after minor revision.

**Response:** We appreciate the reviewer's thoughtful review and constructive comments. All the comments have been addressed in the revised manuscript, and the responses to each comment are given below.
* * *
**Specific comments**

**Comments:** Ln.89. "*The new scheme is based on the particle linear polarization ratio in AERONET Version 3 aerosol inversion product, which is considered a better indicator for non-spherical 90 dust particles (Shin et al., 2018, 2019)*"

Depolarization ratio in AERONET is calculated with the model of randomly oriented spheroids. There are many challenges in application of this model to the dust particles and accuracy of depolarization calculation is the subject of discussions (e.g. Gasteiger, J., Wiegner, M., Groß, S., Freudenthaler, V., Toledano, C., Tesche, M., and Kandler, K.: Modeling lidar-relevant optical properties of complex mineral dust aerosols, Tellus B, 63, 725-741, 2011). I think corresponding comment should be added to the manuscript.

**Response:** Thank you for the reviewer's reminder. The spheroid shape model may indeed raise errors in particle linear depolarization ratio for mineral dust according to the results of a modeling study by Gasteiger et al. (2011). Even though, dust is the predominant particle to trigger significant depolarized signal, which can be well captured by AERONET spheroid model. We take a large threshold of particle depolarization ratio (0.30) to mitigate the error introduced by the spheroid model, in terms of the wrong characterization of dust. We have added some sentences to mention the spheroid model of AERONET retrieval and discuss the benefit of using irregular particle shape in the modeling study by Gasteiger et al. (2011) in the introduction of the revised manuscript as follows

**'…It should be noted that the particle linear depolarization ratio values in AERONET retrieval are calculated from the combination of the particle size distribution and complex refractive index based on a spheroid light scattering model (Dubovik et al., 2006). Based on a modeling study, Gasteiger et al. (2011) found that the lidar-measured particle linear depolarization ratio values for pure**

**mineral dust can be better reproduced by using an irregular particle shape assumption compared with using the spheroid shape assumption. Nevertheless, we consider it adequate to adopt AERONET-derived particle linear depolarization ratio values to qualitatively identify the presence of dust in the atmospheric column (Noh et al., 2017).'** (Please see Line 99-105)

**Comments:** Table 1. Uncertainties of estimation is really important point. In estimation of dust backscattering depolarization ratio of smoke is assumed 0.05. But actually it varies in 0.04-0.09 range (Burton et al., 2013), though provided uncertainty 10-30% looks reasonable. However, lidar ratio of dust may vary in 30 sr-60 sr range, so uncertainty of extinction calculation should be higher, but authors provide 15-25% range. Uncertainty of mass concentration should be even higher, but in Table 1 the range 20-30% is given. I think these uncertainties should be clarified.

**Response:** Thank you for pointing out this issue. Considering this reviewer's comments together with the comments from RC3, we have reevaluated the uncertainties for the parameters provided and updated them as seen in the revised Table 1. Now, the uncertainties in the dust backscatter coefficient and dust extinction coefficient should be approximately <49% and <59%. Thus, we consider the uncertainties in $M_d$, $n_{250,d}$, $s_d$, and $s_{100,d}$ are estimated to be approximately <60%. The final uncertainties in INPC and CCNC are still estimated to be <500% and <200% because the largest uncertainty is still contributed by CCN and INP parameterization schemes.

**Reference:**

Burton, S. P., Ferrare, R. A., Vaughan, M. A., Omar, A. H., Rogers, R. R., Hostetler, C. A., and Hair, J. W.: Aerosol classification from airborne HSRL and comparisons with the CALIPSO vertical feature mask, Atmos. Meas. Tech., 6, 1397–1412, https://doi.org/10.5194/amt-6-1397-2013, 2013.

**Comments:** Ln 125. "a good proxy for dust-related CCN concentration $n_{CCN,ss,d}$  " I think should be explained, why "ss" used in subscript.

**Response:** We have added the following sentence **'Here, the subscript 'ss' denotes the water supersaturation.'** (Please see Line 141)

**Comments:** Eq.3. Notations look a bit strange for me. For example, authors use $N_{250,j}$ and in the right part it becomes $n_{250,d,j}$. Why index "d" is absent in the left part?

**Response:** Thank you for pointing out this. We confirm that the subscript 'd' than denotes 'dust aerosol type' should be present in the left part of Eq. (3), (4), and (5). These Eqs. have been modified accordingly. (Please see Line 149-151)

**Comments:** Eq.6. What is $c_d$?

**Response:** $c_{100,d}$ is a CCN-relevant conversion factor. Together with $\chi_d$, $c_{100,d}$ can be used to calculation $n_{100,d}$ with the following equation $n_{CCN,ss,d}(z) = f_{ss,d} \times n_{100,d}(z)$ in Table 1.

**Comments:** Ln 195. Ok, here authors start discussion of spheroids model. But may be better to do it in introduction.

**Response:** We have added some related discussions on the spheroids model of the AERONET depolarization ratio in the introduction. Please see the response to the first specific comment above.

**Comments:** Eq.8 actually repeats Eq.1.

**Response:** We agree with the reviewer's comment that Eq. (8) is similar to Eq. (1). Here we still give Eq. (8) because column-integrated PLDR data at 1020 nm from AERONET aerosol inversion are used to obtain the column-integrated dust ratio ($R_{d,1020}$), which is somewhat different from the application in height-resolved lidar retrieval. Thus, we would like to retain Eq. (8) for clarity.

---

## Author Comment (AC4)

**Response to RC3**

**General Comments:**

The paper "POLIPHON conversion factors for retrieving dust-related cloud condensation nuclei and ice-nucleating particle concentration profiles at oceanic sites" presents and discusses the dust-related conversion factors as extracted over remote oceanic/coast sites using the AERONET database around the globe. These different conversion parameters are of critical importance for the POLIPHON methodology in order to compute dust-related CCNC and INPC globally. The study falls within the scope of AMT. The authors have done a thorough job, the manuscript is well-written / structured, the presentation clear, the language fluent and the quality of the figures high. Furthermore, the authors give credit to related work and the results support the conclusions. However, in order to help improving the manuscript, I would kindly suggest the authors to take into account the following minor comments.

**Response:** We appreciate the reviewer's thoughtful review and constructive comments. All the comments have been addressed in the revised manuscript, and the responses to each comment are given below.
* * *
**Specific comments:**

**Comments:** Central component of the analysis provided is the AERONET-based depolarization ratio, which is established according to the model of randomly oriented spheroids. Thus, I would suggest to the authors to dive into significantly more discussion and details on the major building component of their approach, including methodology, approach, assumptions, and accuracy.

**Response:** Thank you for the reviewer's reminder. The spheroid shape model may indeed induce errors in particle linear depolarization ratio for mineral dust according to the results of the modeling study from Gasteiger et al. (2011). Even though, dust is the predominant particle to trigger significant depolarized signal, which can be well captured by AERONET spheroid model. We take a large threshold of particle depolarization ratio (0.30) to mitigate the error introduced by the spheroid model, in terms of the wrong characterization of dust. We have added some sentences to mention the spheroid model of AERONET retrieval and the benefit of using irregular particle shape for mineral dust in the modeling study by Gasteiger et al. (2011) in the introduction of the revised manuscript as follows

**'…It should be noted that the particle linear depolarization ratio values in AERONET retrieval are calculated from the combination of the particle size distribution and complex refractive index based on a spheroid light scattering model (Dubovik et al., 2006). Based on a modeling study, Gasteiger et al. (2011) found that the lidar-measured particle linear depolarization ratio values for pure mineral dust can be well reproduced by using an irregular particle shape**

**compared with using the spheroid shape assumption. Nevertheless, we consider it adequate to adopt AERONET-derived particle linear depolarization ratio values to qualitatively identify the presence of dust in the atmospheric column (Noh et al., 2017).'** (Please see Line 99-105)

Besides, in the methodology part (the third paragraph of section 2.2), we have already mentioned this issue. (Please see Line 211-218)

**Comments:** Since a significant number of PollyXT lidars operate at same time AERONET stations, my suggestion would be the extended intercomparison and evaluation of the AERONET-based depolarization ratio against the Polly lidar depolarization ratio, under events of dust, polluted dust, and non-dust, in order to strengthen the argument of the suitability of the AERONET-based depolarization ratio to extract CCNC and INPC conversion factors. This comparison over land should be a first stepping stone before attempting over ocean, where lidar systems are less frequently operated, and eventually before the claim of supporting 3D CCNC and INPC dust-related studies globally.

**Response:** Thank you for the suggestion. It would be better to fully confirm the validity of AERONET-derived PLDR with those measured by ground-based polarization lidar. As we mentioned in the manuscript, the related comparisons have been made and the results have been discussed in many previously published papers, such as Toledano et al. (2019), Shin et al. (2017), and Müller et al. (2010, 2012), especially for the comparisons in SALTRACE (Saharan Aerosol Long-range Transport and Aerosol–Cloud-Interaction Experiment) campaign at Barbados (Haarig et al., 2022), where is a great location to compare the PLDR values from AERONET and lidar measurements in transatlantic dust cases.

**Comments:** The authors should go into more details on the dependencies between the AERONET-based depolarization ratio to extract CCNC and INPC conversion factors around the globe and the discrepancies in dust microphysical properties of dust around the globe, for the main objective is to apply the conversion factors eventually in lidar observations through POLIPHON, possible at regions and conditions of dust transport significantly different than the observed at the specific stations of the present study. Moreover, the authors should discuss, possible through study cases, the change of the extracted and proposed CCNC and INPC conversion factors as a function of aeolian transport and distance, for aging and mixing with non-dust aerosol subtypes, even under the hypothesis of external mixing, alters the columnar observations of AERONET, thus affects the total conversion factors.

**Response:** We are appreciated for the reviewer's valuable suggestion. As mentioned in this manuscript, there will be a following-up work that focuses on the dust-related conversion factors at those polluted city sites with more complicated local aerosol

emission conditions and then the global conversion factor dataset can be expected. AERONET sites are serried in some regions (especially the regions with large populations) and can be very sparse in those outlying regions. Thus, the retrieved conversion factors absolutely will be distributed unevenly. When selecting the AERONET sites for calculating the conversion factors, we try our best to ensure that there are at least regional-representative conversion factors available for most of the geographical locations around the world. A dust-related conversion factor at an isolated site can be applied to a large area around it. Also, geographical interpolation is another possible way to obtain the final global grid dataset of dust-related conversion factors. Nevertheless, the final processing for retrieving the dust-related global conversion factors will be determined only after finishing the following-up work (i.e., conversion factors at polluted city sites), which would be better discussed in detail in our next paper. Therefore, the final global dataset of dust-related conversion factors can reflect the regional characteristics of dust microphysical properties, such as for dust sources, places along dust transport pathways, downstream regions after long-range transport, or regions favoring dust aging and mixing with non-dust aerosols.

However, as the first step, this manuscript focuses on discussing the possibility of a dust case selection scheme employing the AERONET-derived PLDR and attempting the application to the retrieval of dust-related conversion factors at the clean oceanic sites (with simple background aerosol conditions). Besides, another attempt for retrieving the dust-related conversion factor (probably mixing dust situations with other aerosol types) at a polluted city site has been demonstrated by He et al. (2021). To concentrate on the main subject, we would like to mention the future work in the outlook part as seen in the last paragraph of the revised manuscript as below **'Once those conversion factors at polluted city sites are retrieved, a global dust-related conversion factor grid dataset will possibly be obtained by geographical interpolation.'** (Please see Line 397-398)

**Comments:** The aforementioned approach should be as robust as possible, for once the conversion factors are extracted and established for the dust CCNC and INPC over a region, the product output should consist a fingerprint of the dust related sources affecting the region as well, interconnecting the dust plumes over the oceanic sites with the dust sources. Towards this, I would suggest the authors to perform a cluster analysis of the dust sources affecting each oceanic site (i.e., backtrajectories).

**Response:** Thank you for the reviewer's suggestion. As mentioned in our response to the next comment (see below), we agree with the reviewer that different oceanic/coast sites in this study may be influenced by long-range transported dust aerosols from different dust sources over the world. However, the purpose of this study is to obtain the multi-year average characteristic of dust aerosols and associated dust-related conversion factors for the selected oceanic/coast AERONET sites. We do not intend to separate the respective contribution of different dust sources to a given site because it would be much more complicated to analyze the dust sources for different sites and

regions, which has already been studied specifically in the existing literature (Bullard et al., 2016; Struve et al., 2020; Kok et al., 2021; Meinander et al., 2022).

In addition, it is believed that robustness can be guaranteed adequately. First, the data durations are long enough for the selected AERONET sites, which are at least 7 years (St_Helena) and can be up to 28 years (Mauna_Loa and Cape_Verde). Second, the linear Pearson correlation coefficients are generally >0.70 (most of them are >0.90), suggesting the INP-relevant properties for each site are well reflected. Third, the intercomparisons with the conversion factors in Ansmann et al. (2019) using a different dust identification scheme are conducted (in section 3.1). Last, the background atmospheric environment at oceanic sites is always clean, indicating that the identified dust cases are less influenced by other aerosol sources; this issue must be handled with more care when retrieving the dust-related conversion factors at other terrestrial sites in the future.

**Comments:** Please discuss the effect on the extracted AERONET-based depolarization ratios and accordingly on the CCNC and INPC conversion factors of different dust regions – with different dust properties (i.e. LR), affecting the same marine site.

**Response:** It would be difficult to comprehensively and quantificationally discuss this issue. Excluding some occasional extreme events (Uno et al., 2009), a given oceanic region is generally impacted by specific dust sources via typical dust transport pathways. In the middle- and low-latitude Atlantic, the primary dust transport pathway is from the Saharan desert in North Africa to the east coast regions of North America (Rittmeister et al, 2017; Yu et al., 2021). In the North Atlantic, it is reported that dust aerosols are mainly from Iceland (Baddock et al., 2017). Dust aerosols in the Arctic mainly come from the high-latitude dust sources in the North Hemisphere (e.g., Alaska, Canada, Denmark, Greenland, Iceland, Svalbard, Sweden, and Russia) (Bullard et al., 2016; Meinander et al., 2022), Arctic local sources (Shi et al., 2022), Asia (Zhao et al., 2022), and North Africa (Shi et al., 2022). As for the dust aerosols over the Pacific, they mainly originate from the Central and East Asia dust sources to North America (Guo et al., 2017; Hu et al., 2019). As for the remaining few oceanic sites in the South Hemisphere, dust aerosols can be related to Australia, New Zealand, Patagonia, and Southern Africa (Bullard et al., 2016; Struve et al., 2020; Kok et al., 2021; Meinander et al., 2022). Thus, as seen in Figure 6, the region-to-regions variations of conversion factors (i.e., $c_{v,d}$, $c_{s,d}$, and $c_{s,100,d}$) can be attributed to the diverse contributions from different dust sources. In addition, as the downstream areas, the possible aging and mixing of dust with other aerosol types during long-range transport may also be responsible for the region-to-region variations of conversion factors (Kim and Park, 2012; Goel et al., 2020).

Therefore, we have added a new paragraph in section 3.2 to address the reviewer's concern. (Please see Line 313-327)

**Comments:** Table 2 provides the available number of data points for total, dust-dominated mixture, and pure dust, in AERONET inversion products. In specific cases the dataset is characterized by a very low number of cases. The authors should discuss on the fail-safes considered in order to guarantee the robustness of the conversion factors extracted, even in the low number of cases AERONET stations. Moreover, please provide at the table for each of the site (Table 2), with the basis AERONET products, for the Total Obs., DDM Obs., and PD Obs. (i.e., AOD+AOD_SD, AE+AE_SD, …). How does the low number of cases affect the uncertainties and confidence of the conversion factors? Please discuss providing additional input where necessary.

**Response:** In Ansmann et al. (2019b), the lowermost number of available data points for calculating the conversion factors is 17 at the Tuscon AERONET site. In this study, we provide INP-relevant ($c_{v,d}$, $c_{250,d}$, $c_{s,d}$, and $c_{s,100,d}$) conversion factors (see tables 3 and 4) with the number of available data points no less than 12 to ensure as many sites as possible can provide dust-related conversion factors with somewhat acceptable reliability. With data point number below 12, the data points may be diverging caused by occasional dust cases, causing very small linear Pearson correlation coefficients R. Here we provide R values for each dust-related INP-relevant conversion factor at each AERONET employed in this study as seen in the following table. It can be seen clearly that most of the conversion factors have corresponding R values exceeding 0.70, except for PD-derived $c_{250,d}$ values at NR and AS (as marked in red in the table), which can guarantee the robustness of the retrieved conversion factors. Therefore, we have added some sentences to discuss this issue (in the second paragraph of section 3.2) as follows

**'We consider the conversion factors with the number of available PD data points ≥ 12 valid (provided in Table 4). Moreover, to guarantee robustness, only the retrieved conversion factors with the linear Pearson correlation coefficient $R$ exceeding 0.70 are considered valid, except for PD-derived $c_{250,d}$ values at NR (R=0.32) and AS (R=0.50) which should especially be handled with care in scientific application.'** (Please see Line 289-292)

| | Site | R for $C_{v,d}$ | | R for $C_{250,d}$ | | R for $C_{s,d}$ | | R for $C_{s,100,d}$ | |
|---|---|---|---|---|---|---|---|---|---|
| | | DDM+PD | PD | DDM+PD | PD | DDM+PD | PD | DDM+PD | PD |
| **North Africa** | CV | 0.97 | 0.97 | 0.94 | 0.94 | 0.76 | 0.78 | 0.98 | 0.99 |
| | DK | 0.96 | 0.96 | 0.94 | 0.94 | 0.71 | 0.74 | 0.94 | 0.97 |
| | IZ | 0.98 | 0.98 | 0.92 | 0.92 | 0.77 | 0.77 | 0.99 | 0.99 |
| **Middle East** | EI | 0.95 | 0.97 | 0.91 | 0.92 | 0.56 | 0.82 | 0.56 | 0.99 |
| | SV | 0.97 | 0.98 | 0.96 | 0.96 | 0.76 | 0.78 | 0.91 | 0.98 |
| | ME | 0.96 | 0.98 | 0.93 | 0.96 | 0.76 | 0.82 | 0.79 | 0.97 |
| **Asia** | DU | 0.98 | 0.98 | 0.95 | 0.96 | 0.84 | 0.83 | 0.94 | 0.98 |

|  | | | | | | | | | |
|---|---|---|---|---|---|---|---|---|
|  | DA | 0.90 | 0.77 | 0.76 | 0.83 | 0.41 | 0.93 | 0.67 | 0.70 |
|  | LA | 0.96 | 0.97 | 0.86 | 0.93 | 0.76 | 0.73 | 0.95 | 0.99 |
| **Pacific** | TA | 0.88 | - | 0.89 | - | 0.87 | - | 0.94 | - |
|  | NR | 0.93 | 0.80 | 0.90 | 0.32 | 0.91 | 0.79 | 0.96 | 0.76 |
|  | MI | 0.92 | 0.91 | 0.96 | 0.97 | 0.95 | 0.97 | 0.97 | 0.98 |
|  | AS | 0.89 | 0.91 | 0.84 | 0.50 | 0.86 | 0.75 | 0.92 | 0.81 |
|  | GA | 0.80 | - | 0.86 | - | 0.89 | - | 0.93 | - |
|  | ML | 0.88 | 0.92 | 0.92 | 0.95 | 0.90 | 0.95 | 0.95 | 0.99 |
| **Pacific Coast** | HU | 0.84 | 1.00 | 0.84 | - | 0.87 | - | 0.93 | - |
|  | OS | 0.86 | 0.90 | 0.75 | 0.98 | 0.66 | 0.79 | 0.76 | 0.87 |
|  | SH | 0.93 | 0.99 | 0.94 | 0.97 | 0.89 | 0.88 | 0.90 | 0.98 |
|  | SI | 0.92 | - | 0.94 | - | 0.89 | - | 0.96 | - |
|  | TR | 0.88 | 0.98 | 0.93 | 0.98 | 0.93 | 0.97 | 0.95 | 0.98 |
| **Atlantic** | AG | 0.98 | 0.99 | 0.98 | 0.98 | 0.95 | 0.97 | 0.99 | 0.98 |
|  | TH | 0.93 | 0.98 | 0.92 | 0.97 | 0.92 | 0.97 | 0.96 | 0.99 |
|  | ST | 0.91 | - | 0.91 | - | 0.95 | - | 0.98 | - |
| **Indian Ocean** | MG | 0.82 | - | 0.83 | - | 0.92 | - | 0.95 | - |
|  | AI | 0.93 | 0.82 | 0.89 | 0.82 | 0.93 | 0.85 | 0.98 | 0.92 |
| **Arctic Ocean** | NA | 0.92 | - | 0.93 | - | 0.76 | - | 0.88 | - |
|  | TL | 0.95 | - | 0.91 | - | 0.91 | - | 0.95 | - |
|  | OP | 0.76 | - | 0.94 | - | 0.87 | - | 0.91 | - |
|  | IQ | 0.74 | | 0.72 | - | 0.84 | - | 0.88 | - |

Moreover, we have added the AOD (at 532 nm) and AE (between 440 nm and 870 nm) in the updated Table 2 as suggested by the reviewer. Note that table 2 is too crowded to give the corresponding standard deviations for $AOD_{532}$ and $AE_{440-870}$.

**Comments:** In table 1 the authors provide the uncertainties of the approach. The uncertainties have been established on the basis of long-term ground-based observations (i.e., EARLINET, PollyNET). Since the objective of the study, as mentioned in the very beginning of the manuscript, is to "to characterize the 3-D distribution of dust-related Cloud Condensation Nuclei Concentration (CCNC) and Ice Nucleating Particle Concentration (INPC) globally", which can be achieved only based on satellite-lidar systems (i.e., CALIOP, CATS, ATLID), where the uncertainties of backscatter and particulate depolarization ratio are of the same order of magnitude as

the backscatter and particulate depolarization ratio. In this case, as the higher uncertainties are used as input in the error propagation, the final uncertainties will be significantly higher in satellite-based lidar products than when ground based products are extracted. Please discuss.

**Response:** Thank you for pointing out this issue. We agree with the reviewer's point of view that the uncertainties in the retrieved aerosol extinction/backscatter coefficient from spaceborne lidar measurement may differ from those for ground-based lidar measurement. Hence, we have used global CALIOP level-2 aerosol profile data during the night on 1 January 2010 as an example, to examine the typical uncertainties in aerosol extinction and backscatter coefficient as well as particle linear depolarization ratio (only select data points with PDR≥0.05) at 532 nm (see the figures below).

[Figure]

Figure 1R. The Cumulative density of relative error/uncertainty of different CALIOP level-2 aerosol profile products: (a) total backscatter coefficient at 532 nm; (b) extinction coefficient at 532 nm; (c) particle linear depolarization ratio at 532 nm. Both of these results were calculated based on CALIPSO night-time orbits on 1 January 2010.

CALIOP level-2 aerosol profile data with the 5-km horizontal resolution are used here, corresponding to 15 laser pulses. In this situation, the uncertainties in the aerosol extinction coefficient, aerosol backscatter coefficient, and are approximately <180% and <120%. As for the particle linear depolarization ratio, the uncertainty is larger; here, we considered it as <300%. Typically, we merge the raw data of ground-based lidar to obtain a time resolution of one minute (this time can be even larger as 15 min or 30 min

are always used in INP retrieval with ground lidar observations), corresponding to 1200 laser pulses (if using a laser with a pulse repetition frequency of 20 Hz). Thus, from 15 laser pulses to 1200 laser pulses, the uncertainty will be declined by a factor of ~9. As a result, we would like to estimate the uncertainty in aerosol extinction coefficient, aerosol backscatter coefficient, and particle linear depolarization ratio to be approximately <20%, <13%, and <33%, respectively. Considering the uncertainty in the dust lidar ratio (30-60 sr) of 33% and the assumed non-dust depolarization ratio (0.05) of 30% (Burton et al., 2013), the uncertainties in the dust backscatter coefficient and dust extinction coefficient should be approximately <49% and <59%. Thus, we consider that the uncertainties in $M_d$, $n_{250,d}$, $s_d$, and $s_{100,d}$ are estimated to be approximately <60%. Similar to the original table 1, the final uncertainties in INPC and CCNC are still estimated to be <500% and <200%. In addition, it should also be noted that the uncertainty level of CALIOP-derived optical parameters can be further improved by integrating the data to decrease the spatio-temporal resolution. However, the largest uncertainty is contributed by the parameterization schemes for CCN and INP currently; hence, the improvement of lidar-derived optical parameters makes no sense for the moment at least.

Thus, we have updated table 1 based on the uncertainties of optical properties for space-borne lidar measurement accordingly.
* * *
**Reference:**

Burton, S. P., Ferrare, R. A., Vaughan, M. A., Omar, A. H., Rogers, R. R., Hostetler, C. A., and Hair, J. W.: Aerosol classification from airborne HSRL and comparisons with the CALIPSO vertical feature mask, Atmos. Meas. Tech., 6, 1397–1412, https://doi.org/10.5194/amt-6-1397-2013, 2013.

Baddock, M., Mockford, T., Bullard, J., and Thorsteinsson, T.: Pathways of high-latitude dust in the North Atlantic, Earth Planet Sc. Lett., 459, 170-182, https://doi.org/10.1016/j.epsl.2016.11.034, 2017.

Bullard, J. E., Baddock, M., Bradwell, T., Crusius, J., Darlington, E., Gaiero, D., Gassó, S., Gisladottir, G., Hodgkins, R., McCulloch, R., McKenna-Neuman, C., Mockford, T., Stewart, H., and Thorsteinsson, T.: High-latitude dust in the Earth system, Rev. Geophys., 54, 447–485, https://doi.org/10.1002/2016RG000518, 2016.

Guo, J., Lou, M., Miao, Y., Wang, Y., Zeng, Z., Liu, H., He, J., Xu, H., Wang, F., Min, M., and Zhai, P.: Trans-Pacific transport of dust aerosol originated from East Asia: Insights gained from multiple observations and modeling, Environ. Pollut., 230, 1030–1039, https://doi.org/10.1016/j.envpol.2017.07.062, 2017.

Haarig, M., Ansmann, A., Engelmann, R., Baars, H., Toledano, C., Torres, B., Althausen, D., Radenz, M., and Wandinger, U.: First triple-wavelength lidar observations of depolarization and extinction-to-backscatter ratios of Saharan dust, Atmos. Chem. Phys., 22, 355–369, https://doi.org/10.5194/acp-22-355-2022,

2022.

He, Y., Zhang, Y., Liu, F., Yin, Z., Yi, Y., Zhan, Y., and Yi, F.: Retrievals of dust-related particle mass and ice-nucleating particle concentration profiles with ground-based polarization lidar and sun photometer over a megacity in central China, Atmos. Meas. Tech., 14, 5939–5954, https://doi.org/10.5194/amt-14-5939-2021, 2021.

Hu, Z., Huang, J., Zhao, C., Ma, Y., Jin, Q., Qian, Y., Leung, L. R., Bi, J., and Ma, J.: Trans-Pacific transport and evolution of aerosols: spatiotemporal characteristics and source contributions, Atmos. Chem. Phys., 19, 12709–12730, https://doi.org/10.5194/acp-19-12709-2019, 2019.

Kok, J. F., Adebiyi, A. A., Albani, S., Balkanski, Y., Checa-Garcia, R., Chin, M., Colarco, P. R., Hamilton, D. S., Huang, Y., Ito, A., Klose, M., Li, L., Mahowald, N. M., Miller, R. L., Obiso, V., Pérez García-Pando, C., Rocha-Lima, A., and Wan, J. S.: Contribution of the world's main dust source regions to the global cycle of desert dust, Atmos. Chem. Phys., 21, 8169–8193, https://doi.org/10.5194/acp-21-8169-2021, 2021.

Meinander, O., Dagsson-Waldhauserova, P., Amosov, P., Aseyeva, E., Atkins, C., Baklanov, A., Baldo, C., Barr, S. L., Barzycka, B., Benning, L. G., Cvetkovic, B., Enchilik, P., Frolov, D., Gassó, S., Kandler, K., Kasimov, N., Kavan, J., King, J., Koroleva, T., Krupskaya, V., Kulmala, M., Kusiak, M., Lappalainen, H. K., Laska, M., Lasne, J., Lewandowski, M., Luks, B., McQuaid, J. B., Moroni, B., Murray, B., Möhler, O., Nawrot, A., Nickovic, S., O'Neill, N. T., Pejanovic, G., Popovicheva, O., Ranjbar, K., Romanias, M., Samonova, O., Sanchez-Marroquin, A., Schepanski, K., Semenkov, I., Sharapova, A., Shevnina, E., Shi, Z., Sofiev, M., Thevenet, F., Thorsteinsson, T., Timofeev, M., Umo, N. S., Uppstu, A., Urupina, D., Varga, G., Werner, T., Arnalds, O., and Vukovic Vimic, A.: Newly identified climatically and environmentally significant high-latitude dust sources, Atmos. Chem. Phys., 22, 11889–11930, https://doi.org/10.5194/acp-22-11889-2022, 2022.

Müller, D., Ansmann, A., Freudenthaler, V., Kandler, K., Toledano, C., Hiebsch, A., Gasteiger, J., Esselborn, M., Tesche, M., Heese, B., Althausen, D., Weinzierl, B., Petzold, A., and von Hoyningen-Huene, W.: Mineral dust observed with AERONET Sun photometer, Raman lidar, and in situ instruments during SAMUM 2006: Shape-dependent particle properties, J. Geophys. Res., 115, D11207, https://doi.org/10.1029/2009JD012523, 2010.

Müller, D., Lee, K.-H., Gasteiger, J., Tesche, M., Weinzierl, B., Kandler, K., Müller, T., Toledano, C., Otto, S., Althausen, D., and Ansmann, A.: Comparison of optical and microphysical properties of pure Saharan mineral dust observed with AERONET Sun photometer, Raman lidar, and in situ instruments during SAMUM 2006, J. Geophys. Res., 117, D07211, https://doi.org/10.1029/2011JD016825, 2012.

Noh, Y., Müller, D., Lee, K., Kim, K., Lee, K., Shimizu, A., Sano, I., and Park, C. B.: Depolarization ratios retrieved by AERONET sun–sky radiometer data and comparison to depolarization ratios measured with lidar, Atmos. Chem. Phys., 17, 6271–6290, https://doi.org/10.5194/acp-17-6271-2017, 2017.

Rittmeister, F., Ansmann, A., Engelmann, R., Skupin, A., Baars, H., Kanitz, T., and Kinne, S.: Profiling of Saharan dust from the Caribbean to western Africa – Part 1: Layering structures and optical properties from shipborne polarization/Raman lidar observations, Atmos. Chem. Phys., 17, 12963–12983, https://doi.org/10.5194/acp-17-12963-2017, 2017.

Shi, Y., Liu, X., Wu, M., Zhao, X., Ke, Z., and Brown, H.: Relative importance of high-latitude local and long-range-transported dust for Arctic ice-nucleating particles and impacts on Arctic mixed-phase clouds, Atmos. Chem. Phys., 22, 2909–2935, https://doi.org/10.5194/acp-22-2909-2022, 2022.

Struve, T., Pahnke, K., Lamy, F., Wengler, M., Böning, P., and Winckler, G.: A circumpolar dust conveyor in the glacial Southern Ocean, Nature Commu., 11, 5655, https://doi.org/10.1038/s41467-020-18858-y, 2020.

Toledano, C., Torres, B., Velasco-Merino, C., Althausen, D., Groß, S., Wiegner, M., Weinzierl, B., Gasteiger, J., Ansmann, A., González, R., Mateos, D., Farrel, D., Müller, T., Haarig, M., and Cachorro, V. E.: Sun photometer retrievals of Saharan dust properties over Barbados during SALTRACE, Atmos. Chem. Phys., 19, 14571–14583, https://doi.org/10.5194/acp-19-14571-2019, 2019.

Uno, I., Eguchi, K., Yumimoto, K., Takemura, T., Shimizu, A., Uematsu, M., Liu, Z., Wang, Z., Hara, Y., and Sugimoto, N.: Asian dust transported one full circuit around the globe, Nature Geosci., 2, 557–560, https://doi.org/10.1038/ngeo583, 2009.

Yu, H., Tan, Q., Zhou, L., Zhou, Y., Bian, H., Chin, M., Ryder, C. L., Levy, R. C., Pradhan, Y., Shi, Y., Song, Q., Zhang, Z., Colarco, P. R., Kim, D., Remer, L. A., Yuan, T., Mayol-Bracero, O., and Holben, B. N.: Observation and modeling of the historic "Godzilla" African dust intrusion into the Caribbean Basin and the southern US in June 2020, Atmos. Chem. Phys., 21, 12359–12383, https://doi.org/10.5194/acp-21-12359-2021, 2021.

Zhao, X., Huang, K., Fu, J. S., and Abdullaev, S. F.: Long-range transport of Asian dust to the Arctic: identification of transport pathways, evolution of aerosol optical properties, and impact assessment on surface albedo changes, Atmos. Chem. Phys., 22, 10389–10407, https://doi.org/10.5194/acp-22-10389-2022, 2022.